# Functional genomics uncovers the transcription factor BNC2 as required for myofibroblastic activation in fibrosis

Tissue injury triggers activation of mesenchymal lineage cells into wound-repairing myofibroblasts, whose unrestrained activity leads to fibrosis. Although this process is largely controlled at the transcriptional level, whether the main transcription factors involved have all been identified has remained elusive. Here, we report multi-omics analyses unraveling Basonuclin 2 (BNC2) as a myofibroblast identity transcription factor. Using liver fibrosis as a model for in-depth investigations, we first show that *BNC2* expression is induced in both mouse and human fibrotic livers from different etiologies and decreases upon human liver fibrosis regression. Importantly, we found that *BNC2* transcriptional induction is a specific feature of myofibroblastic activation in fibrotic tissues. Mechanistically, BNC2 expression and activities allow to integrate pro-fibrotic stimuli, including TGFβ and Hippo/YAP1 signaling, towards induction of matrisome genes such as those encoding type I collagen. As a consequence, *Bnc2* deficiency blunts collagen deposition in livers of mice fed a fibrogenic diet. Additionally, our work establishes BNC2 as potentially druggable since we identified the thalidomide derivative CC-885 as a BNC2 inhibitor. Altogether, we propose that BNC2 is a transcription factor involved in canonical pathways driving myofibroblastic activation in fibrosis.

Fibrosis is characterized by the excessive accumulation of structurally abnormal extracellular matrix (ECM) that can lead to disruption of tissue architecture and function, and ultimately organ failure[1]. As such, fibrosis has a major clinical impact on patients' morbidity and mortality. This pathological condition is a consequence of an excessive wound healing response caused by repeated or chronic injuries that can affect a wide range of organs, including the kidneys, the lungs, the heart, and the liver[1,2]. Cirrhosis is the terminal stage of progressive liver fibrosis, which has various etiologies. While the incidence of fibrosis caused by viral hepatitis is declining, fibrosis triggered by a spectrum of liver diseases ranging from alcoholic-related liver diseases (ARLD) to non-alcoholic steatohepatitis (NASH) is on the rise. In particular, NASH is rapidly becoming one of the leading causes of liver failure worldwide with respect to the ever-increasing prevalence of obesity[3,4]. While fibrotic-related diseases have been estimated to account for as much

as 45% of all-cause mortality rates in developed countries[5], few therapeutic strategies are currently available and the medical need for effective anti-fibrotic therapies is high[2,5]. As of today, only bariatric surgery has been able to trigger significant liver fibrosis regression in a cohort of patients with obesity-associated NASH[6]. In this context, a better knowledge of the molecular mechanisms involved in fibrosis development is urgently needed.

Despite being initially triggered in an organ- and condition-specific manner, fibrosis converges toward conserved core cellular and molecular pathways[1,7]. A central and common event in fibrogenesis is the unrestrained activation of local tissue fibroblasts or mesenchymal-like cells into highly contractile and ECM-producing myofibroblasts (MFs)[1,8]. Myofibroblastic activation is triggered by intercellular cross-talks within inflamed injured organs where transforming growth factor (TGF)-β is considered the most potent pro-fibrotic cytokine[9].

✉e-mail: jerome.eeckhoute@inserm.fr

Importantly, mechanosensing of the microenvironment stiffness through the Hippo/Yes-Associated Protein 1 (YAP1) pathway has recently emerged as a central trigger of both initial and self-sustained myofibroblastic activation[10,11]. Indeed, expression of ECM encoding genes and ECM-associated genes such as modifying enzymes, together defined as the matrisome[12], allows MFs to increase both ECM amount and stiffness[13]. In particular, MFs massively secrete fibrillar collagens such as type I and III collagens (COL-I and COL-III) and promote their stabilization through cross-linking by lysyl-oxidases (LOX)[9,13].

Myofibroblastic activation relies on a profound transcriptomic reprogramming allowing cells often considered as "quiescent" in healthy conditions to adopt the MF phenotype. This has triggered a growing interest in defining the transcriptional regulators involved in this process. In that regard, despite reported tissue specificities, MFs harbor conserved functionalities across organs such as ECM production pointing to a potentially conserved transcriptional regulatory program. This is exemplified by the described role for TGFβ-induced SMAD transcription factors and for the YAP1 co-regulator in MFs from diverse tissues[8]. In this study, we have implemented an unbiased functional genomics approach to assess candidate transcription factors (TFs) driving the MF phenotype. Our study pointed to Basonuclin 2 (BNC2), an understudied TF that belongs to the Zinc-finger family[14], as specifically expressed in MFs with a predicted role in ECM synthesis. In line, we have found that myofibroblastic activation requires BNC2 up-regulation and that BNC2 expression is a characteristic of MFs from mouse and human fibrotic organs. We have also defined how BNC2 integrates profibrotic signals to directly drive the transcriptional induction of matrisome genes in MFs, and finally we have shown that, as a consequence, BNC2 plays a key role in ECM deposition during liver fibrogenesis. Altogether, our study has uncovered BNC2 as a core TF driving myofibroblastic activation in fibrosis.

## Results

### Functional genomics analyses identify BNC2 as a MF identity TF

Leveraging epigenomic data has recently been established as a robust mean to identify genes defining the identity and specific functionalities of a given cell type[15,16]. In particular, identity genes are characterized by broad histone H3 lysine 4 trimethylated (H3K4me3) domains around their transcriptional start site (TSS), a feature functionally linked to enhanced transcriptional activity and consistency[15,17]. Hence, in order to identify key genes for MF identity and function, we mined H3K4me3 chromatin immunoprecipitation experiments coupled to DNA sequencing (ChIP-seq) data from human primary MFs, obtained by spontaneous in vitro activation of hepatic stellate cells (HSCs)[18], the main source of MFs in the liver[19,20]. MFs obtained upon in vitro activation lack induction of environmentally induced inflammatory genes, but the transcriptional program responsible for the cell-intrinsic and canonical MF functions is appropriately induced[8,21–23]. Chromatin regions enriched for H3K4me3 signal in MFs were identified and split into sharp and broad H3K4me3 domains (Fig. 1A and Supplementary Fig. 1A). As expected, promoters characterized by broad H3K4me3 domains were more active as indicated by wider H3K27ac ChIP-seq signals around their TSS (Supplementary Fig. 1B) and higher mRNA expression levels of associated genes (Supplementary Fig. 1C) when compared to genes with sharp H3K4me3 domains. Also, promoters with broad H3K4me3 were characterized by shorter distances to nearest super-enhancers (Supplementary Fig. 1D), another defining feature of identity genes[24]. Among genes labeled with broad H3K4me3 domains, we have previously defined that identity genes are best captured by broad H3K4me3 domains which are not ubiquitously conserved across cell types[25]. Indeed, conserved broad H3K4me3 domains rather point to genes, such as tumor suppressor genes, involved in essential cell functions[25,26]. Hence, in order to more specifically

discriminate potential identity genes, well-annotated genes linked to broad MF H3K4me3 domains were retrieved ($n = 1278$), and length of H3K4me3 labeling at these genes was monitored in 76 other human cell types and tissues[26]. Hierarchical clustering was next used to sort both genes and samples according to the relative length of gene H3K4me3 domains in the different analyzed cell types. This allowed to identify 3 main gene clusters denoted as clusters 1, 2, and 3, comprising 130, 318, and 830 genes, respectively (Fig. 1B and Supplementary Data file 1). Cluster 1 stood out as comprising genes whose H3K4me3 domain length was remarkably and preferentially high in a cluster of samples comprising MFs together with MF-like cells such as in vitro grown mesenchymal cells and muscle stem cells[27] (Fig. 1B and Supplementary Data File 2). Consistently, mining transcriptomic data from 561 primary human cells (Supplementary Data File 3) indicated that MFs were characterized by high expression of cluster 1 genes. Indeed, when cells were independently ranked according to the average expression of genes from clusters 1, 2, and 3, a biased ranking of MFs towards highly expressing cells was evidenced for cluster 1 (Fig. 1C). Moreover, pathway and gene ontology enrichment analyses revealed that, while genes from clusters 2 and 3 were specifically enriched for essential cell functions or signaling pathways, genes from cluster 1 were more significantly enriched for terms related to acquisition of a mesenchymal phenotype including ECM synthesis and organization, i.e., MF canonical functions (Fig. 1D and Supplementary Fig. 1E). In line, cluster 1 comprised the greatest proportion of matrisome genes (Fig. 1E). Altogether, these data pointed to cluster 1 genes as particularly relevant with regards to the MF molecular identity.

In order to define potential key transcriptional regulators operating in MFs, we screened the list of MF identity genes from cluster 1 for the presence of TFs. 25 TFs were identified, among which several had already been described to play a role in the control of myofibroblastic activation or had been linked to fibrosis, including Forkhead Box (FOX), T-Box (TBX), Small Mothers Against Decapentaplegic (SMAD), TWIST and SNAIL family members (Fig. 2A). Importantly, our strategy also pointed out TFs more seldomly or never previously reported to be important in MFs, including the poorly studied zinc-finger TFs called Zinc Finger Homeobox 4 (ZFHX4) and Basonuclin 2 (BNC2). Further supporting these predictions, we found BNC2 expression levels to be high in MFs in general regardless of their tissue of origin (Fig. 2B). In line, BNC2 expressing cells also showed higher expression of the MF signature genes Actin Alpha 2, Smooth Muscle (ACTA2 also known as α-SMA) and Collagen Type I Alpha 1 Chain (COL1A1) (Fig. 2B). Moreover, Gene-Module Association Determination annotation (G-MAD), which predicts gene functions by analyzing co-expressed genes across large-scale transcriptomic data from human samples[28], pointed out genes linked to ECM-related functions when interrogated using BNC2 (Fig. 2C). These predicted functional relationships hold true for many different human organs when they were individually considered (Supplementary Fig. 2A). While ZFHX4 was also largely expressed across MFs (Supplementary Fig. 2B), G-MAD scores linking ZFHX4 to ECM were not as strong across different human organs (Supplementary Fig. 2A). Moreover, similar analyses using TBX4, a previously described lung MF-specific TF (Supplementary Fig. 2A, C, D)[29], further highlighted the strong and widespread association of BNC2 with ECM-related terms (Supplementary Fig. 2A). Finally, in addition to being characterized by a broad H3K4me3 domain and H3K27ac super-enhancer, we found that the BNC2 gene was also densely bound by the acetylated histone epigenetic reader BRD4 for its expression in MFs (Supplementary Fig. 3A). Consequently, induction of BRD4 degradation with MZ1 led to reduced BNC2 levels (Supplementary Fig. 3B, C). This is in line with BNC2 defining an identity TF since expression of identity TF genes is characterized by its sensitivity to BRD4 inhibition[25,30].

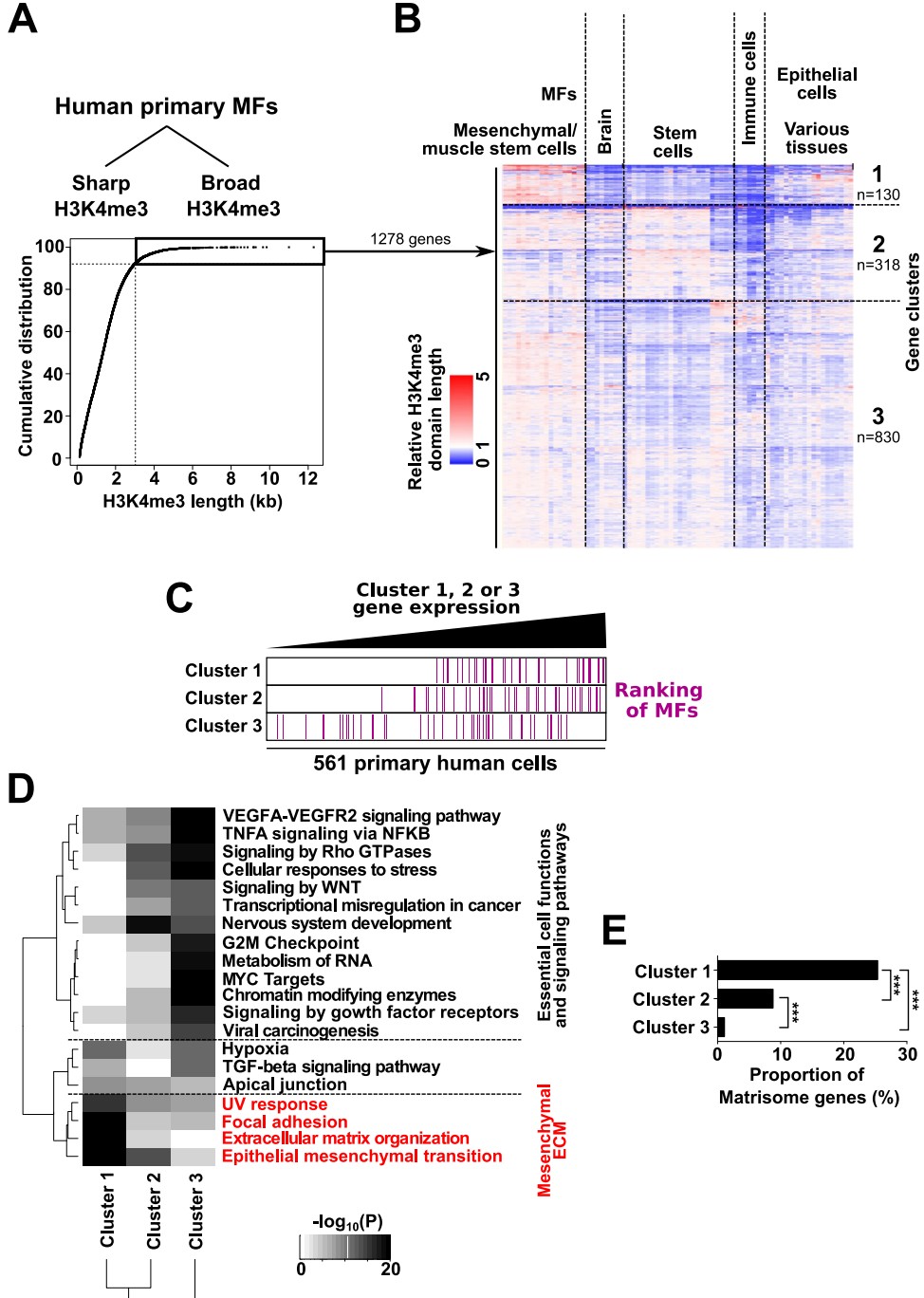

**Fig. 1 | Identification of MF identity genes through large-scale epigenomic and transcriptomic analyses. A** Breadth distribution of H3K4me3 domains called from ChIP-seq data from human primary MF-HSCs. The cumulative frequency distribution of H3K4me3 domain lengths is shown. The inflexion point of the curve was used to separate sharp from broad H3K4me3 domains. **B** Heatmap showing the relative length of H3K4me3 domains in 76 human cell types or tissues for 1278 genes linked to a broad H3K4me3 domain in primary MFs identified in (**A**). Hierarchical clustering was used to cluster both genes (rows) and samples (columns). Clusters were visualized on the heatmap using dotted lines. Three main gene clusters were identified and labeled clusters 1–3 (Supplementary Data File 1). At the top of the heatmap, the identity of cell types or tissues within the 5 main sample clusters are summarized (details are provided in Supplementary Data File 2). **C** 561 primary human cells were ranked based on

increasing median expression levels of genes from cluster 1, 2, or 3. Next, the position of 64 MF cell types was indicated by purple horizontal bars within the ranking obtained using genes from cluster 1, cluster 2 or cluster 3. **D** Top 20 terms retrieved by Metascape[88] when searching for pathways enriched within genes from clusters 1–3 (Hallmark Gene Sets, Reactome Gene Sets, KEGG Pathway, WikiPathways, Canonical Pathways, and PANTHER Pathways were used). Hierarchical clustering based on statistical significance is shown. White color indicates a lack of significance. Terms colored in red are preferentially enriched in cluster 1. **E** Proportion of matrisome genes in clusters 1–3. The percentage of matrisome genes in each cluster was retrieved by comparing genes from clusters 1–3 with a list of 1028 genes encoding ECM and ECM-associated proteins[12]. Statistical significance was assessed using two-sided Fisher exact tests with Benjamini–Hochberg correction for multiple testing.

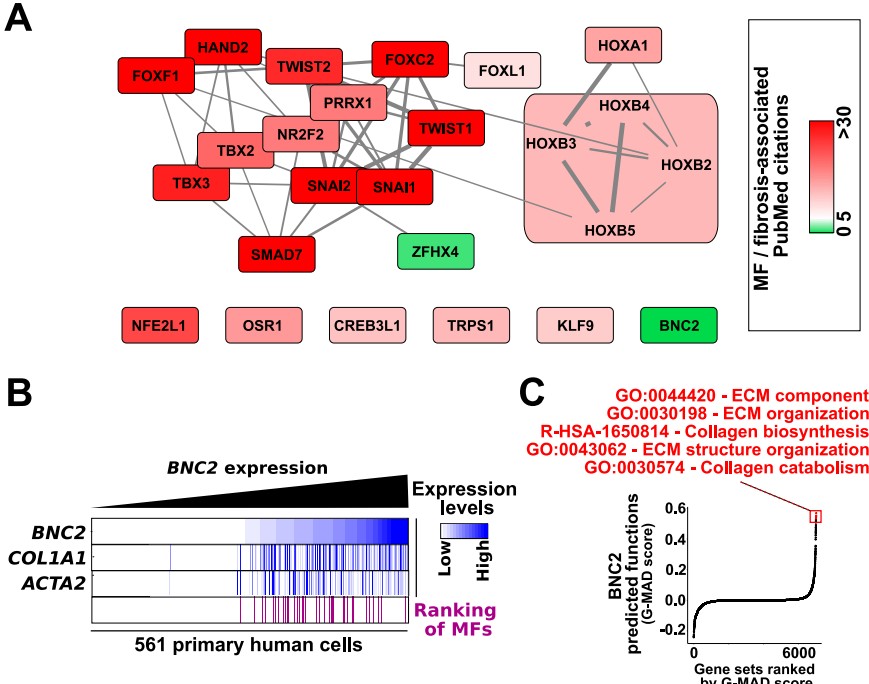

**Fig. 2 | Identification of BNC2 as a MF identity TF. A** Network-based presentation of the 25 TFs identified by our approach. The connection between TFs was provided by String[108]. TFs are colored according to the number of PubMed citations linking a given TF to MFs or fibrosis as detailed in the Methods section. TFs of the HOXB family were grouped together. TFs in the green background have no reported link to MFs or fibrosis. **B** Analysis performed as in (**C**) except cells were ranked according to increasing *BNC2* expression levels. *COL1A1* and *ACTA2* expression levels are displayed using the same color scale. Ranking of 64 MF cell types is indicated at the bottom using purple horizontal bars. **C** Prediction of BNC2-associated functions by the Gene-module association Determination (G-MAD) tool. The top five terms (highest G-MAD scores) are shown.

Altogether, these data prompted us to perform an in-depth characterization of the role exerted by BNC2 in myofibroblastic activation and fibrosis.

## Induction of BNC2 expression specifically characterizes myofibroblastic activation in fibrotic tissues

To characterize BNC2 expression with regards to myofibroblastic activation, we performed in-depth analyses of BNC2 expression upon both murine and human liver fibrogenesis. Increased *Col1a1* expression was used throughout these experiments as a molecular marker of fibrosis. We found that *Bnc2* expression was increased in mouse fibrotic livers obtained through repeated injection of carbon tetrachloride ($CCl_4$) for 8 weeks, a classical model of hepatotoxicity-induced liver fibrosis (Fig. 3A and Supplementary Fig. 4A). Importantly, this induction specifically stemmed from myofibroblastic hepatic stellate cells (MF-HSCs). Indeed, sorting the main liver cell types (hepatocytes (Hep), Kupffer cells (KC), HSC, cholangiocytes (Chol), and liver sinusoidal endothelial cells (LSEC)) from $CCl_4$-injected mice showed that *Bnc2* gene expression was specifically induced in MF-HSCs (Fig. 3B). In line, single-cell RNA-seq analyses of the livers of $CCl_4$ treated mice[31] indicated that *Bnc2* transcripts were specifically found in MF-HSCs expressing the highest levels of *Col1a1* (Supplementary Fig. 5A–D). Of note, *BNC2* induction upon in vivo myofibroblastic activation extended to lung or heart injury (Supplementary Fig. 5E) in line with *BNC2* being a marker of MFs across organs. In the liver, *Bnc2* expression was also induced when mice were subjected to diet-induced fibrosis, i.e., high fat, sucrose and cholesterol diet (HFSC)[32] and choline-deficient L-amino-acid-defined HFSC (CDAA-HFSC) diet[33] (Fig. 3C, D and Supplementary Fig. 4B, C). Single-nuclei RNA-seq analysis of liver cells from mice fed a HFSC diet[34] further pointed to *Bnc2* being expressed in MFs (Supplementary

Fig. 6). Hence, *Bnc2* expression was consistently induced in different mouse liver fibrosis models.

To monitor BNC2 expression in human livers, we first used samples from a cohort of patients with alcohol-related cirrhosis (Supplementary Data File 4)[35]. RT-qPCR assays showed that *BNC2* expression is significantly induced in human cirrhotic livers when compared to histologically normal areas of livers from patients who underwent tumor resections (Fig. 4A). In line, BNC2 protein expression levels were specifically detected in livers with alcohol-related cirrhosis together with that of type I collagen (COL-I) and ACTA2 (Fig. 4B; see Supplementary Fig. 7 for anti-BNC2 specificity controls). Importantly, RNAscope mRNA in situ hybridization performed on four human cirrhotic livers indicated that *BNC2* and *COL1A1* mRNA expression co-localize to the fibrous tissue, while their expression is low within the hepatocyte areas and in non-fibrotic control livers (Fig. 4C and Supplementary Fig. 8A).

Interrogating our previous transcriptomic analyses[36] indicated that *BNC2* is also significantly upregulated in human NASH-associated fibrosis (Supplementary Fig. 8B and Supplementary Data file 5). Since liver fibrosis resolution involves loss of MF-HSCs through different mechanisms, including MF-HSC apoptosis, senescence, or deactivation[37], we hypothesized that fibrosis resolution would be linked to a loss of *BNC2* expression. To test this hypothesis, we leveraged liver biopsies obtained from a long-term follow-up study of a subset of patients with NASH-associated fibrosis who underwent bariatric surgery resulting in liver fibrosis regression (Kleiner score F3 at the time of surgery to Kleiner score F0/1 when monitored 1–5 years after surgery)[6]. Interestingly, liver fibrosis regression obtained through bariatric surgery was accompanied by reduced *BNC2* expression levels as demonstrated by RNAscope assays in four patients (Fig. 4D and Supplementary Fig. 8C).

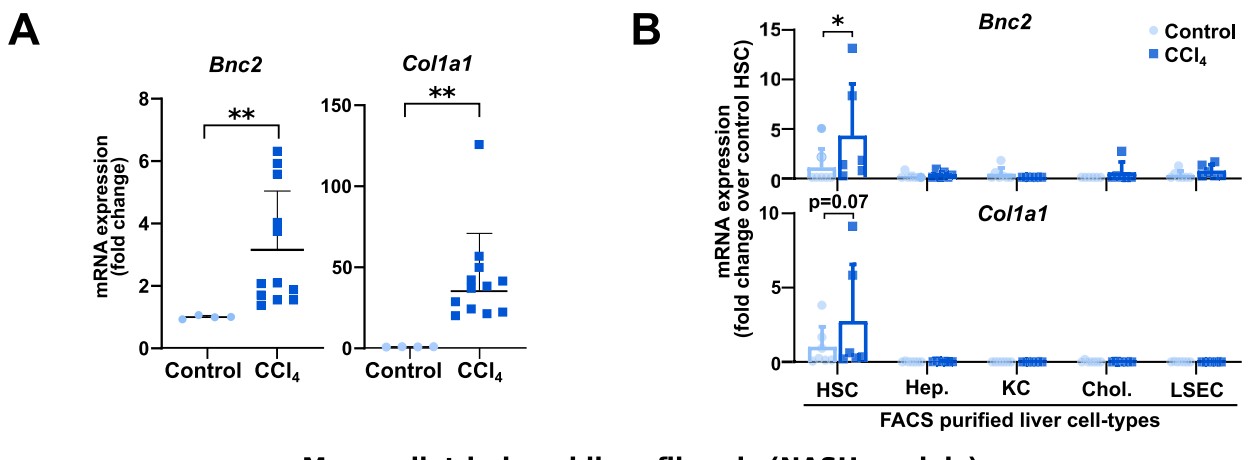

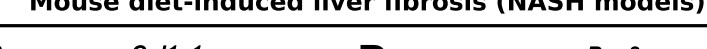

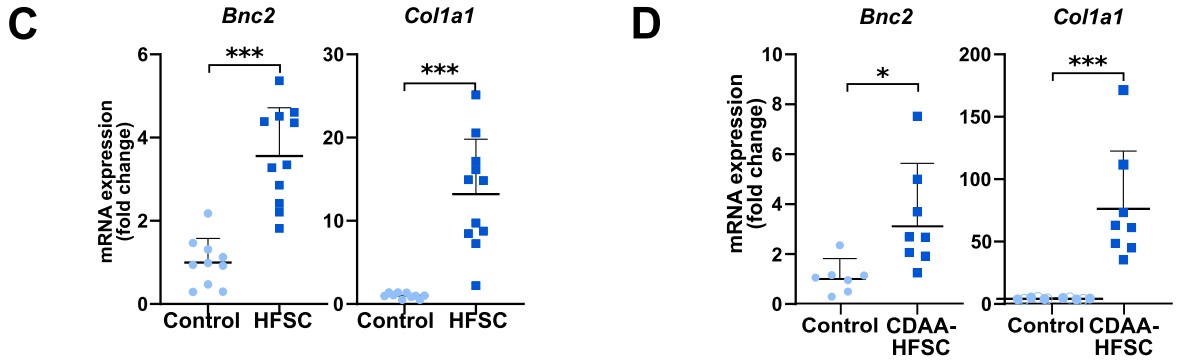

**Fig. 3 | Induction of *Bnc2* expression characterizes MF-HSCs in mouse liver fibrosis. A** RT-qPCR data showing increased *Bnc2* (*n* = 12 mice) and *Col1a1* (*n* = 11 mice) expression in the fibrotic livers of mice injected with CCl₄. Control mice were injected with olive oil (*n* = 4 mice). **\*\****P* = 0.011. **B** RT-qPCR analysis of *Bnc2* and *Col1a1* expression in FACS-sorted cells obtained from the livers of mice injected with CCl₄ (*n* = 6 mice) or olive oil (*n* = 7 mice). Hep. hepatocytes, KC Kupffer cells, HSC hepatic stellate cells, Chol. cholangiocytes, LSEC liver sinusoidal endothelial cells. **\****P* = 0.0111. **C** RT-qPCR data showing increased *Bnc2* and *Col1a1* expression in

the livers of mice fed the HFSC diet for 24 weeks (*n* = 11 mice) or a standard rodent chow diet (control, *n* = 10 mice). **\*\*\****P* < 0.0001. **D** RT-qPCR data showing increased *Bnc2* and *Col1a1 mRNA* expression in the livers of mice fed the CDAA-HFSC diet for 10–12 weeks (*n* = 8 mice) or a standard rodent chow diet (control, *n* = 7 mice for *Bnc2* and 6 for *Col1a1*). **\****P* = 0.0289 and **\*\*\****P* = 0.0007. Graphs in panels **A**–**D** show means ± SD. Statistical significance was assessed using two-tailed Mann–Whitney *U* test (**A** and **C**, **D**) or two-way ANOVA with Sidak multiple comparison post hoc test (**B**, CCl₄ vs control). Source data are provided as a Source Data file.

Altogether, these data indicated that induction of BNC2 expression is an intrinsic feature of myofibroblastic activation which characterizes mouse and human liver fibrosis.

## BNC2 integrates pro-fibrotic stimuli in MF-HSCs

In order to define the functional role exerted by BNC2 in MFs, a combination of cistromic and epigenomic analyses were performed in the human MF-HSC cell line LX2. ChIP-seq was used to define the genome-wide binding sites of BNC2 using an antibody qualified through rapid immunoprecipitation mass spectrometry of endogenous protein (RIME) experiments[38] as being able to immunoprecipitate cross-linked BNC2 (Supplementary Data File 6). We identified 1479 BNC2 binding sites, which occurred in active transcriptional regulatory regions as judged through measurement of chromatin accessibility using Column Purified chromatin (CoP)-seq[39] and histone acetylation using H3K27ac ChIP-seq (Fig. 5A and Supplementary Data File 7). Enrichment for H3K27ac at identified BNC2 binding sites also specifically extended to primary human MF-HSCs (Supplementary Fig. 9A). De novo motif discovery retrieved the previously proposed BNC2 consensus binding motif[40] as the top enriched motif (Supplementary Fig. 9B). To further mine the BNC2 cistrome, we next used the CistromeDB ChIP-seq database toolkit[41] to monitor overlaps with known cistromes. Interestingly, transcriptional regulators involved in the fibrogenic TGFβ (SMAD) and Hippo/YAP1 (TEAD/YAP1) signaling

pathways were identified in these analyses (Fig. 5B). Accordingly, DNA motifs used by the SMAD or TEAD TF families are enriched within the BNC2 cistrome (Supplementary Fig. 9B, C). This is in line with previous observations indicating that the TGFβ and Hippo/YAP1 signaling pathways converge in the regulation of myofibroblastic activation[8,42] and shows this occurs at BNC2-bound regulatory regions. Indeed, recruitment of SMAD3 and YAP1 (ChIP-seq data from LX2 or IMR90 cells)[43,44] is an overall feature of active regulatory elements bound by BNC2 in MFs (Fig. 5C). In line, RIME assays used to define BNC2 chromatin-bound partners in LX2 cells retrieved several cofactors involved in signal transduction from stiff ECM including YAP1 (Fig. 5D, Supplementary Fig. 10, and Supplementary Data Files 6 and 8). A physical interaction between BNC2 and the co-regulator YAP1 was confirmed by co-immunoprecipitation experiments (Fig. 5E and Supplementary Fig. 9D). Identification of genes bound by BNC2 using the Genomic Regions Enrichment of Annotations Tool (GREAT)[45] pointed to ECM-related genes such as *COL1A1, COL5A1,* or *Lysyl Oxidase Like 1* (*LOXL1*) (Fig. 5F and Supplementary Fig. 9E). BNC2 binding to these genes was verified using additional ChIP-qPCR assays (Supplementary Fig. 9F). Interestingly, concomitant recruitment of BNC2, SMAD3, and YAP1 was also observed at the *BNC2* gene itself (Fig. 5G). This was consistent with identity TFs being involved in auto-regulatory loops[46] and suggested that induction of BNC2 expression upon myofibroblastic activation could be driven by the TGFβ and Hippo/

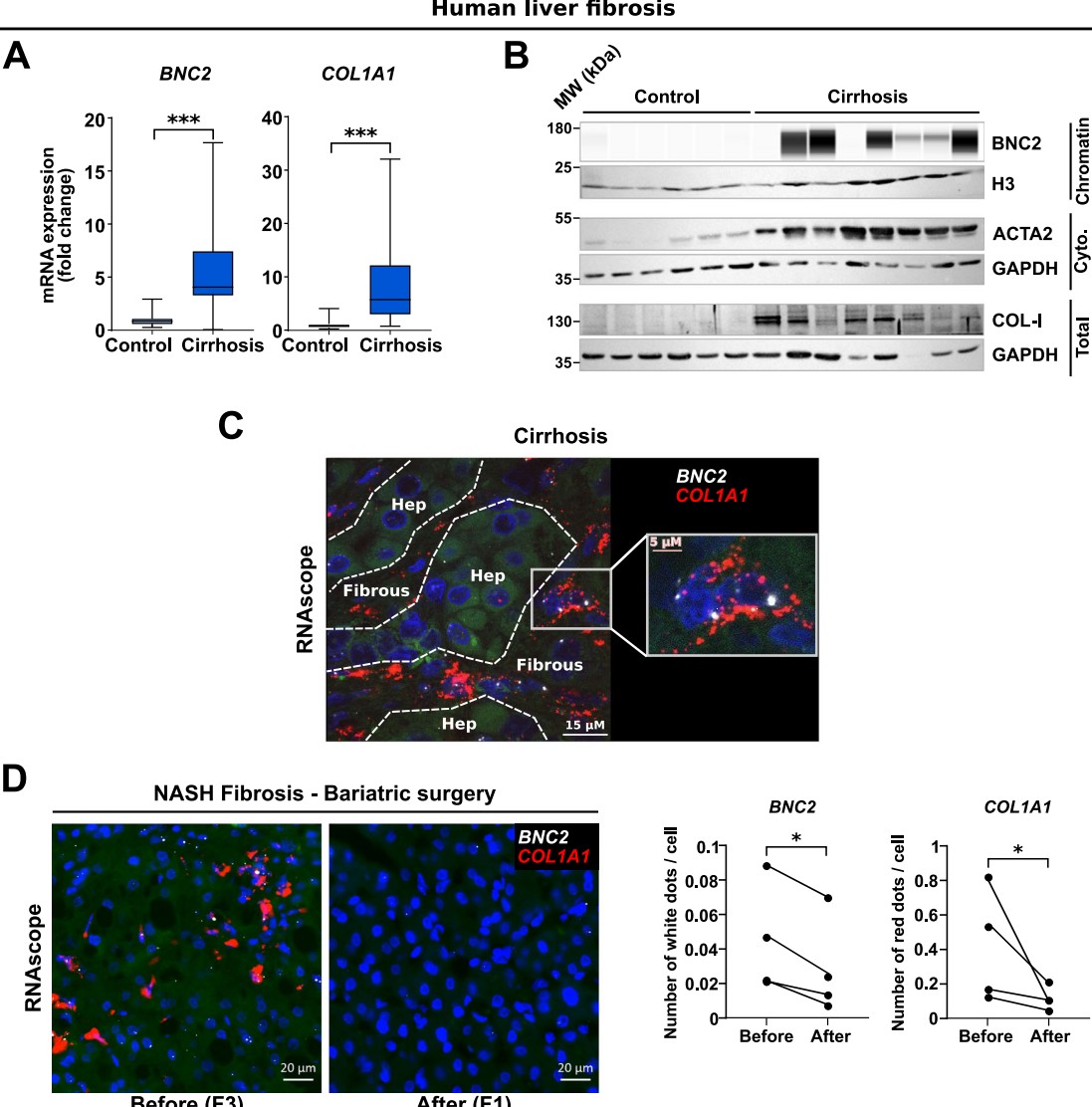

**Fig. 4 | Induction of BNC2 expression characterizes MF-HSCs in human liver fibrosis. A** RT-qPCR data showing increased *BNC2* and *COL1A1* expression in the livers of patients presenting alcohol-related cirrhosis (*n* = 42 biologically independent samples) *versus* control biopsies from the nontumoral areas of liver resections (*n* = 20 biologically independent samples). Box plots are composed of a box from the 25th to the 75th percentile with the median as a line and min to max as whiskers. Statistical significance was assessed using two-tailed Mann–Whitney *U* test. ***P* < 0.0001. **B** Western blot assays showing BNC2, ACTA2 and type I Collagen (COL-I) protein expression in the livers of patients with alcohol-related cirrhosis (= 8 biologically independent samples) and control samples from the nontumoral areas of liver resections (= 6 biologically independent samples). Sub-cellular fractionation was performed for the detection of BNC2 (chromatin fraction) and ACTA2 (cytosolic fraction). H3 and GAPDH were used as loading controls. MW molecular weight markers. **C** Localization of *BNC2* (white staining) and *COL1A1* (red staining) mRNA in human alcohol-related cirrhotic livers. In situ RNA hybridation (RNAscope) was performed on paraffin-embedded liver sections obtained from patients presenting alcohol-related cirrhosis (*n* = 4 biologically independent samples) or control samples from the nontumoral areas of liver resections (*n* = 2 biologically independent samples). Dotted lines delimitate the hepatocyte and fibrous areas. **C** shows a representative image obtained from the liver of a patient with alcohol-related cirrhosis (Patient #A). Data obtained with livers from additional donors are shown in Supplementary Fig. 8A. **D** RNAscope analyses of *BNC2* and *COL1A1* mRNA expression in liver biopsies from patients with obesity-associated NASH showing bariatric surgery-induced loss of liver fibrosis (Kleiner Score F3 before surgery to F0/F1 1–5 years after surgery; *n* = 4 biologically independent sets of samples). **D** shows representative images obtained from the liver of one patient (Patient #G). Data obtained with livers from additional donors (*n* = 3) are shown in Supplementary Fig. 8C. The number of dots/cell at baseline and after surgery are shown in the right panel. Statistical significance was assessed two-tailed ratio paired *t* test. *P* = 0.0416 or 0.0426 for *BNC2* and *COL1A1*, respectively. Source data are provided as a Source Data file.

YAP1 signaling pathways. Indeed, induction of *Bnc2* expression was recapitulated upon spontaneous in vitro activation of purified primary quiescent murine HSCs (Q-HSCs) into MF-HSCs (Fig. 6A and Supplementary Fig. 11A, B), which involves mechanotransduction through the Hippo/YAP1 pathway due to the high stiffness substrate provided by plastic culture plates[8,47]. Furthermore, treatment with the YAP1 inhibitor verteporfin[48] and transfection of a small interfering RNA (siRNA) targeting *Yap1* (siYap1) led to downregulation of *Bnc2* expression in in vitro activated primary murine MF-HSCs (Fig. 6B, C). Moreover, growing primary murine Q-HSCs in non-embedded 3D-spheroids, which reduces YAP1 activation[47], was also accompanied by lower *Bnc2* expression levels when compared to conventional 2D cultures (Fig. 6D). In addition, we found that *Bnc2* expression in MF-HSCs was further induced by treatment with TGFβ (Fig. 6E) and inversely reduced by treatment with SB431542 or SD208, which are TGFβ receptor kinase inhibitors (Fig. 6F).

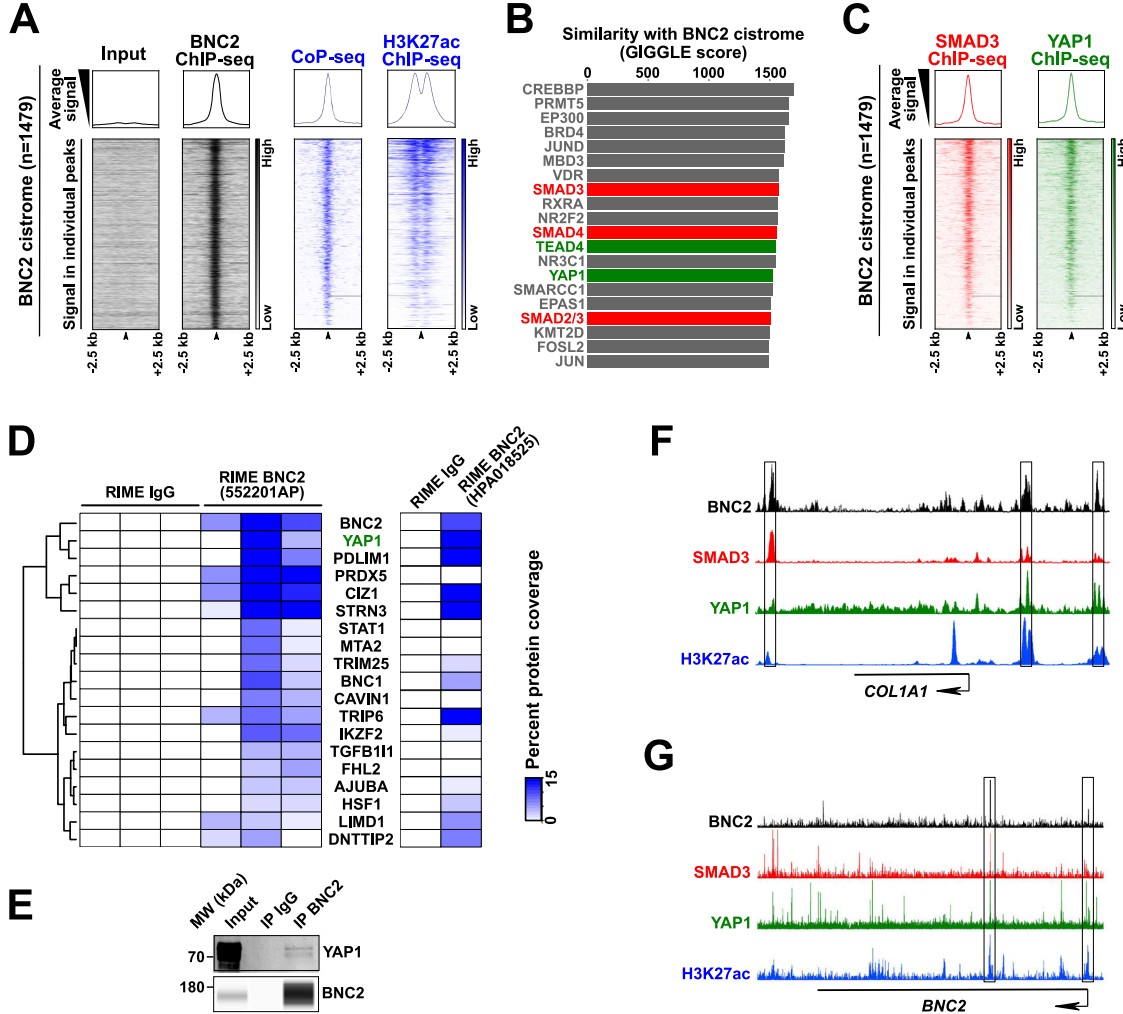

**Fig. 5 | BNC2 cistrome and interactome point to functional intermingling with key fibrogenic signaling pathways. A** Input and BNC2 ChIP-seq signals together with CoP-seq and H3K27ac ChIP-seq signals from LX2 cells were visualized at the BNC2 cistrome (1479 binding sites). Heatmaps at the bottom show the signals in 5 kb regions centered on the BNC2 peaks. Average signals are plotted on top of the heatmaps. **B** The BNC2 cistrome was used to search for proteins with similar genomic binding profiles in CistromeDB and the top 20 hits were ranked according to the GIGGLE similarity score. Proteins involved in TGFβ signaling (SMADs) and Hippo/YAP1 (TEAD and YAP1) were highlighted in red and green, respectively. **C** Analyses performed as in A to monitor SMAD3 and YAP1 ChIP-seq signals from MFs at the BNC2 cistrome. **D** Transcriptional regulators specifically identified in BNC2 RIME but not IgG control RIME were clustered according to the percent

protein coverage in individual biological replicates (*n* = 3 biologically independent experiments; anti-BNC2 antibody 55220-1-AP, Proteintech). Percent coverage obtained in an additional BNC2 RIME experiment using the anti-BNC2 antibody HPA018525 (Sigma-Aldrich) is shown on the right. **E** Nuclear extracts from LX2 cells expressing recombinant BNC2 and YAP1 were subjected to immunoprecipitation with an antibody against BNC2 (55220-1-AP, Proteintech). Immunoprecipitated material was analyzed by western blot using antibodies directed against BNC2 or YAP1. The presented data are representative of two biologically independent experiments. MW, molecular weight markers. **F, G** The Integrated Genome Browser (IGB) was used to visualize ChIP-seq profiles for BNC2 (black track, LX2 cells), SMAD3 (red track, LX2 cells), YAP1 (green track, IMR90 cells), and H3K27ac (blue track, MF-HSCs) at the *COL1A1* (**F**) and *BNC2* (**G**) genes.

---

Hence, fibrogenic signaling pathways converge at BNC2 binding sites including those found at the *BNC2* gene itself, defining BNC2 as an integrator of fibrogenic stimuli in MF-HSCs.

## BNC2 transcriptionally drives the expression of matrisome genes in MF-HSCs

We next more broadly characterized potential BNC2 target genes. In addition to the previously mentioned assignation of BNC2-bound regions to proximal genes using GREAT, we also performed potential target gene assignation taking into account promoter-enhancer interactions predicted by the FDR-corrected OLS with Cross-validation and Shrinkage (FOCS) tool[49]. These complementary procedures retrieved, as top hits, genes enriched for similar biological processes as revealed by gene ontology term enrichment analyses (Supplementary Fig. 12A). These biological processes were further grouped based on gene-set similarity, which indicated that predicted

BNC2 target genes were enriched for functions directly relevant to MF activities such as "response to wounding" and "ECM" (Fig. 7A, Supplementary Fig. 12B, and Supplementary Data File 9). We next combined these data with transcriptomic analyses of LX2 cells transfected with a siRNA targeting *BNC2* (siBNC2; silencing efficiency is shown in Fig. 8A). Gene Set Enrichment Analysis (GSEA)[50] indicated that most biological processes predicted to be targeted by BNC2 using its cistrome were indeed enriched for deregulated (i.e., up- or down-regulated) genes in BNC2-deficient LX2 cells (Fig. 7A). To further mine BNC2 predicted targets, we monitored enrichment for molecular function gene sets among genes defined as both bound and regulated by BNC2 in these analyses. This notably highlighted an enrichment for genes related to ECM, cell adhesion and signaling such as the TGFβ/SMAD signaling pathway (Fig. 7B, Supplementary Fig. 12C, and Supplementary Data File 9). In line, GSEA indicated that the matrisome gene set was significantly biased towards downregulated genes upon

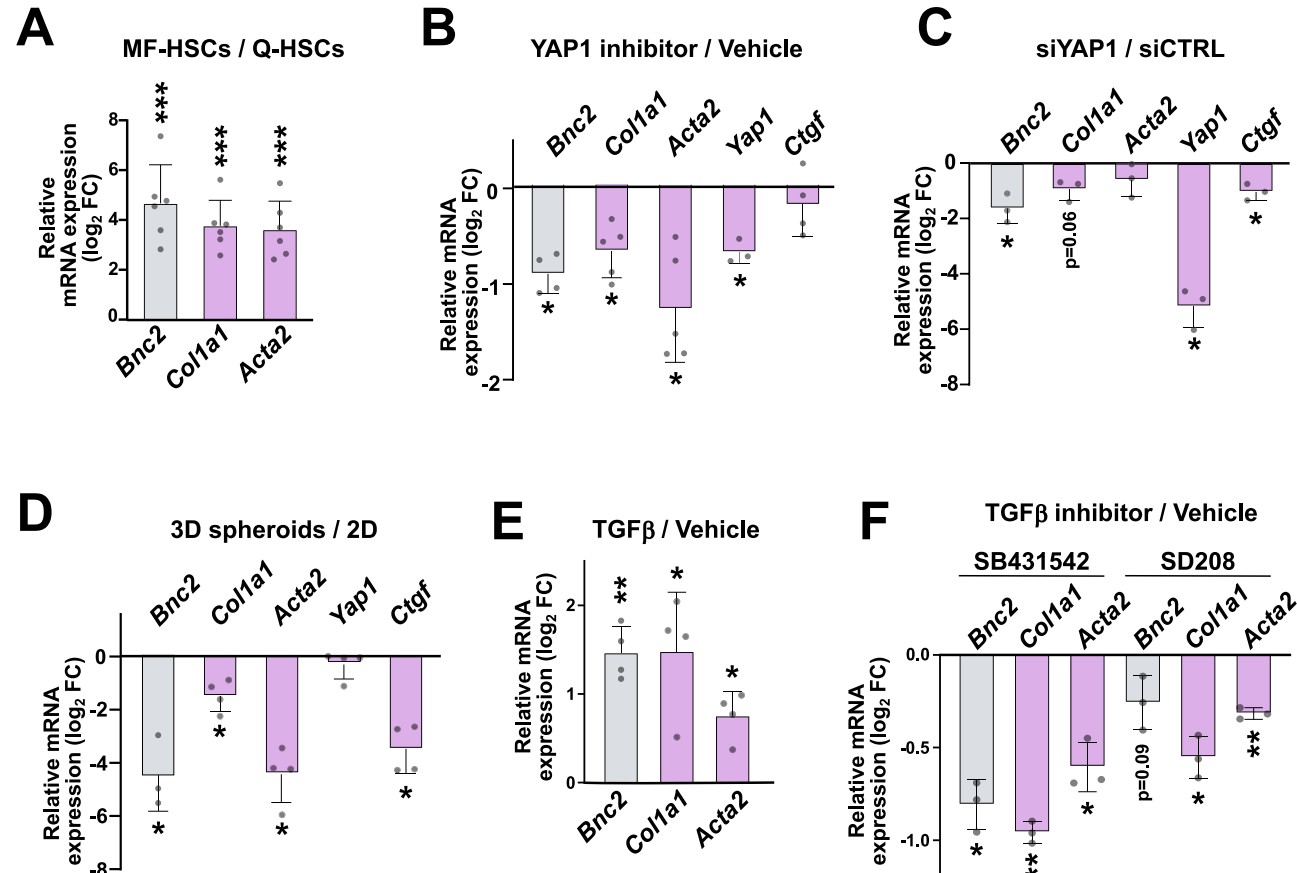

**Fig. 6 | BNC2 expression is induced by profibrogenic signaling pathways in MF-HSCs. A** RT-qPCR data showing changes in gene expression upon murine primary Q-HSCs spontaneous in vitro activation into MF-HSCs (*n* = 6 biologically independent experiments). Log₂ fold changes (FC) between MF-HSCs (6 days of culture) and Q-HSCs (1 day of culture) are shown. **B** RT-qPCR data showing changes in gene expression upon YAP1 inhibition in MF-HSCs. Isolated mouse primary HSCs were treated with 1 μM verteporfin or vehicle for 6 days before being harvested [*n* = 3 (*Yap1*), 4 (*Bnc2* and *Ctgf*), or 5 (*Col1a1* and *Acta2*) biologically independent experiments]. Log₂ FC between verteporfin and vehicle-treated cells are shown. **C** RT-qPCR data showing changes in gene expression upon *Yap1* silencing in MF-HSCs (*n* = 3 biologically independent experiments). Log₂ FC between siYAP1 and siCTRL-transfected cells are shown. **D** RT-qPCR data showing differences in gene expression in mouse primary HSCs grown for 9 days in 3D (spheroids) or in 2D [*n* = 3

(*Bnc2*) or 4 (*Col1a1*, *Acta2*, *Yap1*, *Ctgf*) biologically independent experiments). Log₂ FC between cells grown in 3D and 2D are shown. **B–D** Expression of *Ctgf*, an established YAP1 target, was assessed. **E** RT-qPCR data showing changes in gene expression induced by treatment of MF-HSCs with TGFβ (1 ng/mL) for 24 h (*n* = 4 biologically independent experiments). Log₂ FC between cells treated with TGFβ and vehicle are shown. **F** RT-qPCR data showing changes in gene expression induced by treatment of EMS404 MF-HSCs with the indicated TGFβ signaling inhibitors for 24 h (*n* = 3 biologically independent experiments). Log₂ FC between cells treated with TGFβ signaling inhibitors and vehicle are shown. In all panels, bar graphs show means ± SD. Statistical significance was assessed using two-sided one-sample *t* test with Benjamini–Hochberg correction for multiple testing to determine if the mean log₂ FC was statistically different from 0. *$P < 0.05$, **$P < 0.01$, ***$P < 0.001$. Source data are provided as a Source Data file.

BNC2 silencing (Fig. 7C). Calling differentially regulated genes upon *BNC2* silencing and performing gene enrichment analyses also retrieved matrisome genes as specifically enriched among downregulated genes (Supplementary Fig. 13). Similar conclusions were reached when GSEA analyses were performed using a restricted set of genes encoding ECM components effectively detected in human fibrotic livers[51] (Fig. 7D). GSEA analyses performed on transcriptomic data obtained upon *Bnc2* silencing in the mouse MF-HSC cell line EMS404 (Supplementary Fig. 14) also pointed to downregulation of ECM-related genes (Fig. 7E, F). Both ECM constituents such as collagen encoding genes and ECM regulators including ECM-remodeling enzymes were downregulated in both LX2 and EMS404 cells (Supplementary Fig. S15). The requirement of BNC2 for expression of matrisome-related genes was verified using RT-qPCR in primary human MF-HSCs. Indeed, similar to LX2 cells, BNC2 silencing in primary human MF-HCSs led to decreased expression of a selected set of matrisome genes encoding ECM constituents (*COL1A1*, additional *COL* genes and *Laminin Subunit Alpha 5* (*LAMA5*)), the ECM receptor

*Integrin Subunit Alpha 1* (*ITGA1*) and the ECM cross-linking enzyme *LOXL1* (Fig. 8A). The importance of BNC2 was conserved across species since its silencing led to significant downregulation of most of these ECM-related genes both in EMS404 cells and in mouse primary MF-HSCs (Fig. 8B). In line with reduced expression of *Col1a1* and *Col1a2*, BNC2 silencing decreased protein levels of COL-I in both LX2 and EMS404 cells (Fig. 8C). Of note, downregulation of matrisome gene expression upon BNC2 silencing was also found in LL-29 MF from human idiopathic pulmonary fibrosis (Supplementary Fig. 16) further substantiating the conserved function of BNC2 in MFs. Since Zinc-finger TFs were recently shown to be targeted to cereblon (CRBN)-mediated protein degradation by thalidomide and its derivatives[52], we used thalidomide and the pomalidomide, CC-885 or Iberdomide derivatives as additional potential means to control BNC2 expression in MFs. We found that BNC2 levels were specifically downregulated by CC-885 in LX2 cells (Fig. 8D and Supplementary Fig. 17A). This was in line with in silico predictions based on docking analyses performed using a structural model of the BNC2 zinc finger-CRBN interaction,

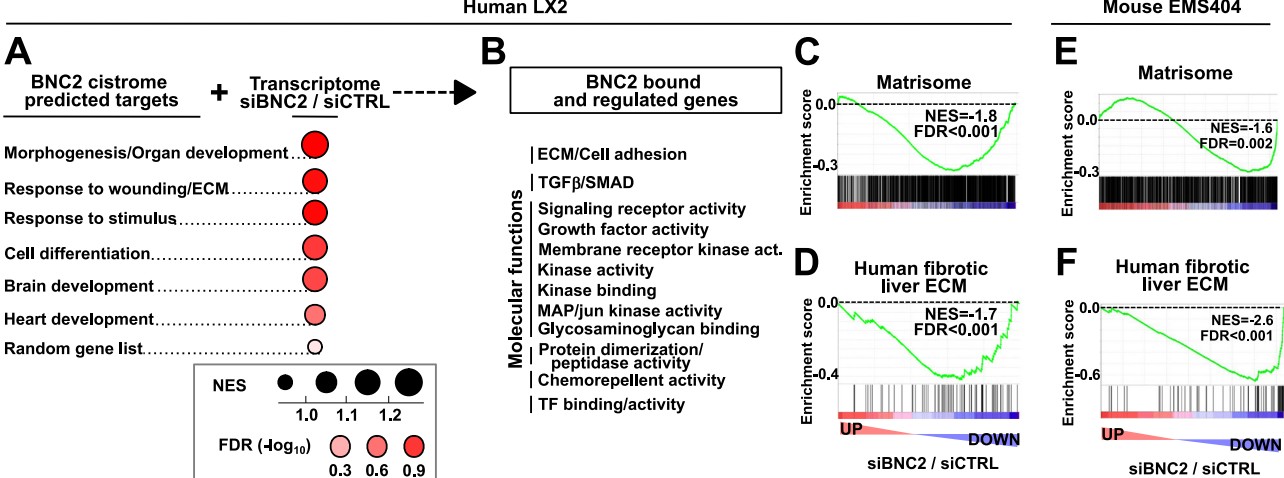

**Fig. 7 | BNC2 controls the MF transcriptome, including fibrogenic ECM-related genes. A** Biological Process GO term enrichment analysis was performed using genes linked to the BNC2 cistrome. The main recovered terms are indicated under "BNC2 cistrome predicted targets". Next, microarray-based analysis was used to define how BNC2 predicted target genes were deregulated upon BNC2 silencing. Dot plot depicts the results of GSEA performed for each term using transcriptomic changes induced by BNC2 silencing in LX2 cells, which were transfected with the BNC2-targeting siRNA (siBNC2) or a non-targeting control siRNA (siCTRL) (*n* = 4 biologically independent experiments). Dot areas are proportional to the normalized enrichment score (NES), while colors indicate the false discovery rate (FDR). A random list of 500 human genes was used as a negative control. **B** Genes both bound and regulated by BNC2, i.e., genes from the GSEA core enrichments from panel **A**, were further mined to search for enriched molecular pathways using ToppGene. Enriched terms were clustered according to the similarity of gene lists (Supplementary Fig. 12C), and main clusters pointing to related common molecular functions were further grouped. **C**, **D** Enrichment plots from GSEA performed using transcriptomic changes induced by siBNC2 in LX2 cells as the ranking measure and the Matrisome (ECM and ECM-associated genes, *n* = 1028; **C**) or human proteomics-based identified fibrotic liver ECM genes (*n* = 71; **D**) as the gene sets. NES and FDR are the normalized enrichment score and the false discovery rate provided by GSEA, respectively. **E**, **F** Enrichment plots from GSEA similar to those shown in (**C**, **D**) obtained using mining of transcriptomic data from EMS404 cells transfected using siBNC2 or siCTRL (control).

which indicated that only CC-885 was able to adopt a bridging position between BNC2 and CRBN which could promote this interaction (Supplementary Fig. 17B). Thalidomide and its derivatives, including CC-885, were ineffective at decreasing BNC2 in mouse EMS404 cells (Supplementary Fig. 17A), in line with mouse cells being resistant to these compounds at least in part because of differences in the CRBN sequence[53]. Interestingly, CC-885 treatment of LX2 cells also selectively led to a decrease in matrisome gene expression (Fig. 8E and Supplementary Fig. 17C).

Collectively, these data define a role for BNC2 in the control of matrisome gene expression in MFs by integrating pro-fibrotic signals through both its expression and its transcriptional regulatory activities.

### BNC2 controls the development of liver fibrosis

In light of our results, we next assessed whether the role for BNC2 in MF-HSCs in vitro translates to a role in fibrosis development in vivo. Ayu21-18 mice, which carry an insertion of a gene trap vector between exons 2a and 3 of the *Bnc2* gene leading to its disruption and inactivation[54,55] were studied. Since *Bnc2* knock-out animals prematurely die within 24 h after birth[55], heterozygous (*Bnc2*[+/−]) mice were used to define whether reduced *Bnc2* expression is sufficient to modulate liver fibrogenesis. The CDAA-HFSC diet was used since it triggers induction of *Bnc2* expression (Fig. 3D) concomitant with moderate NASH-associated liver fibrosis. *Bnc2*[+/−] mice and their wild-type (WT; *Bnc2*[+/+]) littermates were fed the CDAA-HFSC diet for 7.5 weeks (Fig. 9A). Hepatic steatosis (Fig. 9B and Supplementary Fig. 18A) and plasma transaminase activities (Fig. 9C and Supplementary Fig. 18B) were not significantly different between *Bnc2*[+/+] and *Bnc2*[+/−] mice. However, a significant decrease in the level of collagen deposition was measured in *Bnc2*[+/−] mice, as determined by Sirius red staining of liver sections (Fig. 9D). *Bnc2*[+/−] mice were also characterized by reduced liver collagen deposition in an independent experiment where mice were fed the CDAA-HFSC diet for an extended period of 12 weeks

(Supplementary Fig. S19A, B). Moreover, liver transcriptomic analyses revealed a significant downregulation of genes relevant to fibrogenesis in *Bnc2*[+/−] mice including inflammation and ECM-related functions, as evidenced using GSEA (Fig. 9E and Supplementary Fig. 18C, D). Consistently, GSEA using the matrisome and the human fibrotic liver ECM gene sets indicated a significant bias towards genes downregulated in livers of *Bnc2*[+/−] mice, i.e stronger expression in the livers of WT mice (Fig. 9F, G). This was further confirmed by RT-qPCR analyses which indicated that reduced *Bnc2* expression in the livers of *Bnc2*[+/−] mice was accompanied by decreased expression levels of several of our previously selected set of BNC2-dependent MF matrisome genes (Fig. 9H and Supplementary Fig. 19C).

Altogether, these data establish that BNC2-mediated control of matrisome genes is also observed in vivo and, consequently, that this TF is required for liver fibrogenesis in mice.

## Discussion

By combining multi-omics analyses in experimental models of myofibroblastic activation and organ fibrosis, our study has allowed to uncover BNC2 as a TF critical for MF identity and fibrogenesis. Our strategy, leveraging epigenomic and transcriptomic features of identity TF genes, allowed us to capture poorly studied TFs such as BNC2. This is in contrast with previous studies relying on the enrichment of TF-recognition motifs and regulatory network reconstruction (e.g., refs. [56,57]). Indeed, many TFs of the Zinc-finger family including BNC2 have been understudied and are consequently not yet represented in databases of known TF-binding motifs[58,59]. As an illustration, motif-based identification of TFs cooperating with the cofactor YAP1 in MFs did not identify BNC2[60]. In that regard, and in line with ref. [14], our study highlights that multi-omics approaches can be leveraged to provide an in-depth characterization of a given TF's functions by defining how this TF bridges signaling pathways and transcriptional regulatory activities. Indeed, we report that BNC2 acts as an integrator of pro-fibrotic stimuli in MFs where the TGFβ and Hippo/YAP1

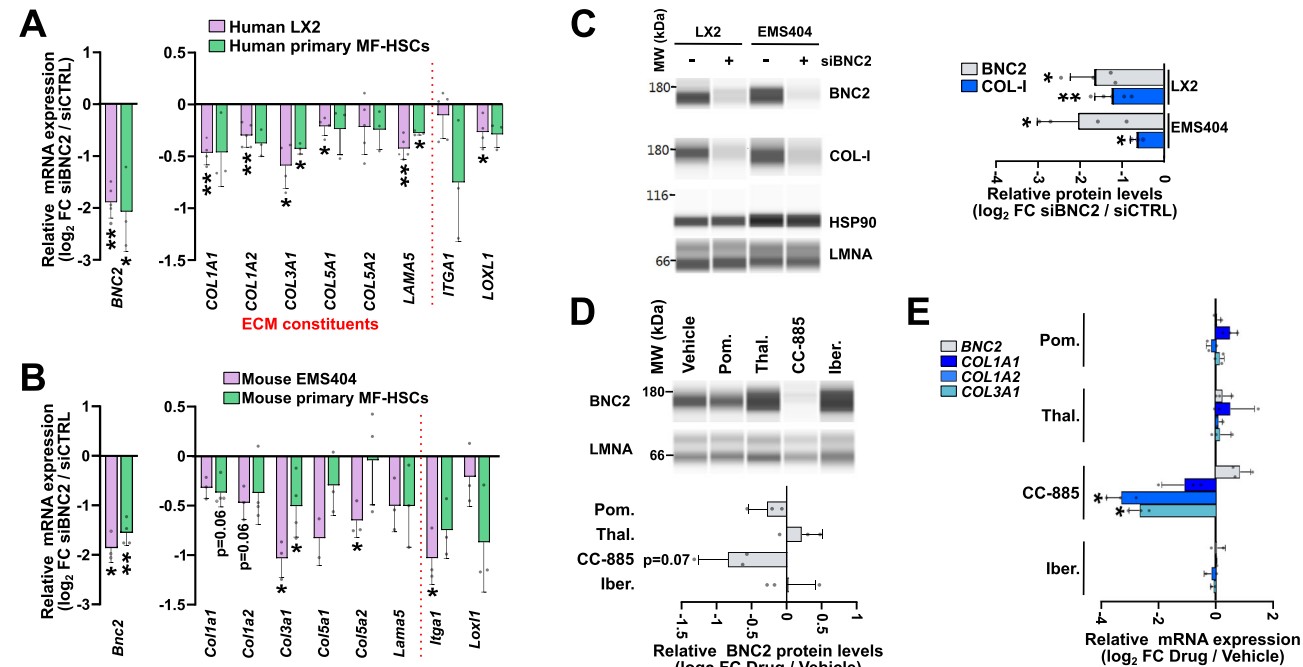

**Fig. 8 | BNC2-mediated regulation of ECM-related gene expression in mouse and human MF-HSCs. A, B** RT-qPCR data showing changes in gene expression upon BNC2 silencing in human (**A**) and mouse (**B**) MF-HSC cell lines and primary cells. Cells were transfected with *BNC2*-targeting siRNA (siBNC2) or a non-targeting control siRNA (siCTRL) for 48 h. Data for LX2 are from biologically independent experiments [$n = 4$ (*COL1A1, COL3A1, COL5A1, LOXL1*) or 5 (*BNC2, COL1A2, COL5A2, LAMA5, ITGA1*)] while data for human primary MF-HSCs were obtained using cells from different donors ($n = 3$). Data for EMS404 are from biologically independent experiments ($n = 3$ or 2 for *Col5a1*) while data for mouse primary MF-HSCs were obtained using cells from independent isolations [$n = 3$ (*Col5a1, Col5a2, Lama5, Itga1, Loxl1*) or 4 (*Bnc2, Col1a1, Col1a2, Col3a1*)]. Log$_2$ FC between siBNC2 and siCTRL-transfected cells are shown. **C** Simple western immunoassays used to monitor changes in BNC2 and type I collagen (COL-I) levels upon BNC2 silencing in LX2 (human) and EMS404 (mouse) cell lines. The lower panel shows the results of protein quantifications obtained from biologically independent experiments [($n = 3$ (COL-I in EMS404), 4 (BNC2), or 5 (COL-I in LX2)]. BNC2 and COL-I levels were

normalized to that of HSP90 or LMNA expression. Log$_2$ FC between siBNC2 and siCTRL-transfected cells are shown. MW, molecular weight markers. **D** Simple Western immunoassays used to monitor changes in BNC2 levels in LX2 cells treated with 1 µM pomalidomide (Pom.), thalidomide (Thal.), CC-885, Iberdomide (Iber.) or vehicle for 24 h. Lower panel shows the results of protein quantification obtained from three biologically independent experiments. Data were normalized to LMNA protein expression, and Log$_2$ FC between drug and vehicle-treated cells are shown. MW, molecular weight markers. **E** RT-qPCR data showing changes in gene expression upon treatment of LX2 cells as described for (**D**) ($n = 3$ biologically independent experiments or 2 for iberdomide). Log$_2$ FC between drug and vehicle-treated cells are shown. Pom., pomalidomide; Thal.; thalidomide; Iber., iberdomide. In all panels, bar graphs show means ± SD. Statistical significance was assessed using two-sided one-sample $t$ test with Benjamini–Hochberg correction for multiple testing to determine if the mean log$_2$ FC was statistically different from 0. *$P < 0.05$, **$P < 0.01$, ***$P < 0.001$. Source data are provided as a Source Data file.

pathways converge both towards the *BNC2* gene itself and BNC2-bound regulatory regions at downstream targets (Fig. 10).

Despite originating from various cell types, MFs across different organs acquire a similar phenotype with canonical functions which include ECM synthesis[9]. Therefore, myofibroblastic activation defines a common theme in fibrogenesis across organs[1]. Our study indicates that this is linked to MFs being characterized by commonalities in their epigenomic and transcriptomic programs. Importantly, we posit that BNC2 is required at the molecular level to establish MF canonical functionalities and more specifically for high expression of ECM constituents and modifying enzymes. In mice, deletion of *Bnc2* is lethal due to the development of a cleft palate and abnormalities of craniofacial bones and the tongue[55]. In humans, *BNC2* mutations have been linked to congenital lower urinary-tract obstruction, adolescent idiopathic scoliosis, and ovarian cancer[40,61,62]. Therefore, we suspect that some of the molecular functions described in our study may also be relevant to these additional BNC2-related phenotypes. Indeed, the developmental functions of BNC2 are linked to its expression in cells of the mesenchymal lineage in the developing craniofacial bones[55]. Moreover, reminiscent of several other MF identity TFs (Fig. 2A; e.g., TWIST, SNAI), BNC2 pro-tumorigenic activities[40,63] may be linked to its ability to promote mesenchymal-like transcriptional programs triggering epithelial–mesenchymal transitions (EMT)[64]. However, beyond this unifying program, single-cell transcriptomic analyses are now also

pointing to MF adopting a myriad of heterogenous phenotypes. In particular, MFs are emerging as instrumental signaling cells in injured organs where adaptation to the local microenvironment may define labile heterogeneous populations of MFs across different stages of the fibrogenic process[65,66]. Our data suggest that BNC2 might also be involved in additional MF activities such as those pertaining to cell signaling and dialog (Figs. 7A, B, 9E). In this context, how BNC2 activities and that of MF subtype-specific TFs are orchestrated will be important to define in order to better understand the molecular mechanisms underpinning MF heterogeneity and plasticity. Indeed, cell identity and fine tuning of the cellular transcriptional program are defined through combinatorial activities of multiple TFs. In that regard, we have obtained evidences that additional TFs identified in our study including the other poorly characterized Zinc-finger TF ZFHX4 also represent bona fide regulators of the MF transcriptional program. Indeed, their silencing in MF-HSCs leads to reduced type I collagen encoding gene mRNA and protein levels (Supplementary Fig. 20).

Fibrosis, which can affect virtually all organs, is a major health issue in desperate need for widely applicable therapeutic treatment options[2]. The ability to reverse fibrosis to avoid life-threatening complications is particularly important but challenging in patients with advanced fibrosis. In this context, targeting myofibroblasts is considered a promising strategy. Considering the high costs of drug

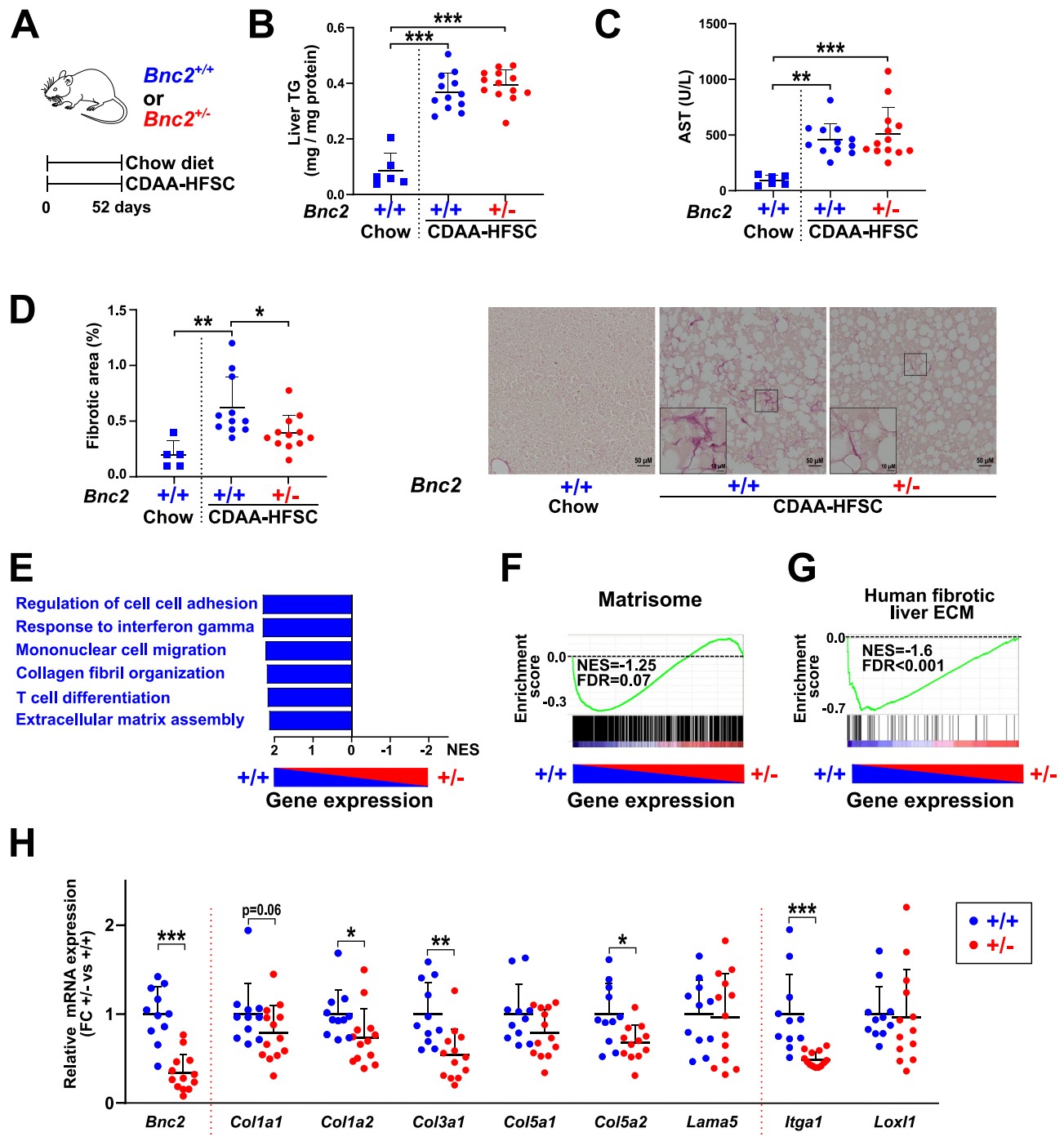

**Fig. 9 | *Bnc2* deficiency dampens induction of matrisome gene expression and collagen deposition by a mouse liver fibrogenic diet. A** Schematic summary of the experimental protocol. Heterozygous mice (+/−, *n* = 13 mice) and WT mice (+/+, *n* = 12 mice) were fed for 7.5 weeks with the CDAA-HFSC diet. Control WT mice were fed a chow diet (*n* = 6 mice). Blood and livers were collected the day of sacrifice for biochemistry, histological, and gene expression analyses shown in panels **B–H**. A mouse was depicted using "Vector diagram of the laboratory mouse (black and white)", https://commons.wikimedia.org/wiki/File:Vector_diagram_of_laboratory_ mouse_(black_and_white).svg, available under the Creative Commons Attribution-Share Alike 4.0 International license (https://creativecommons.org/licenses/by-sa/4. 0/deed.en). **B** Liver triglycerides (TG) content. **C** Aspartate aminotransferase (AST) plasma levels. **D** Quantification of collagen deposition was performed using Sirius red staining of two liver sections per mouse. Ten fields were randomly chosen within each section. The average percentage of fibrotic area is shown. The right panel shows representative images obtained for WT (+/+; *n* = 11 mice) or heterozygous (+/−; *n* = 12 mice) mice. **E** GSEA was used to define biological processes

enriched for genes deregulated in the livers of *Bnc2* heterozygous mice analyzed in (**B–D**). Transcriptomic analyses were performed using livers from ten heterozygous (+/−) and ten WT (+/+) mice. Transcriptomic changes induced by *Bnc2* deficiency were used as the ranking measure and biological processes (MiSigDB, v7.2) as the gene sets. Normalized enrichment scores (NES) for terms with FDR < $10^{-4}$ are shown. **F, G** Enrichment plots from GSEA performed using transcriptomic changes induced by *Bnc2* deficiency in mice as the ranking measure and the Matrisome (ECM and ECM-related genes, *n* = 1028; **F**) or human fibrotic liver ECM genes (*n* = 71; **G**) as the gene sets. NES and FDR are the normalized enrichment score and the false discovery rate provided by GSEA, respectively. **H** RT-qPCR data showing the expression of matrisome genes in WT (+/+; *n* = 11 mice) and heterozygous (+/−; *n* = 13 mice) mice. **B–D, H** Graphs show means ± SD. Statistical significance was assessed using one-way ANOVA with Tukey multiple comparison post hoc test (**B–D**) or one-tailed Mann–Whitney *U* test with Benjamini–Hochberg correction for multiple testing (**H**). *$P < 0.05$, **$P < 0.01$, ***$P < 0.001$. Source data are provided as a Source Data file.

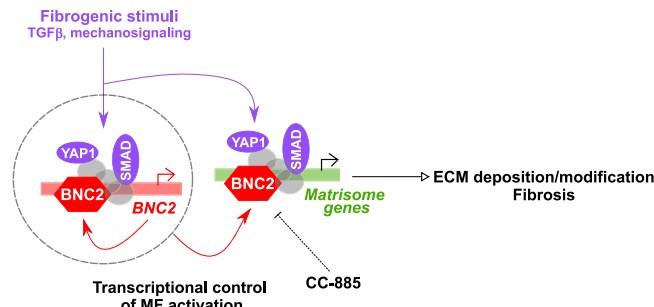

**Fig. 10 | Summary of the main findings of this study.** Schematic depicting how BNC2 controls the establishment of the MF transcriptome and development of fibrosis.

development, the interest of targeting myofibroblasts also lies in their conserved role as drivers of fibrosis development across organs[1,5,7]. Hence, identification of TFs such as BNC2 which drive the common MF identity program and whose expression decreases upon fibrosis regression in humans is of primary interest. While developing drugs targeting TFs outside the nuclear receptor family still represents a tremendous challenge[67], our observation that BNC2 expression levels and MF activities are sensitive to the thalidomide derivative CC-885 suggests this TF could be a potential actionable target in fibrosis.

## Methods

### Resources and reagents
All resources and reagents used in this study are listed in Supplementary Data File 10.

### Animal studies
Mice were housed in standard cages in a temperature-controlled room (22-24 °C) with a 12-h dark–light cycle. Wild-type (WT) C57BL/6J mice were purchased from Charles River and were allowed to acclimate for at least 1 week prior to any experiment. The Ayu21-18 mouse line (B6;CBA-Bnc2Gt(pU21)18Imeg/Orl) was purchased from Infrafrontier (EM-05043). In this model, insertion of the gene trap vector pU21 in the intron separating exons 2a and 3 leads to disruption and inactivation of the *Bnc2* gene[55]. For genotyping, DNA was extracted from the tail with REDExtract-N-Amp Tissue PCR Kit (Sigma-Aldrich) and PCR was performed as described in ref. 55 using the primers listed in Supplementary Data File 10. Primers pairs allowing to detect the WT allele or the mutated allele (directed against the pU21 vector) were mixed in the same PCR (35 repetitions of the following amplification cycle were used: 95° for 15 s, 55° for 15 s, and 72° for 30 s). Amplicons were visualized after migration in ethidium bromide-stained agarose gels as shown in Supplementary Fig. 21. Experiments were performed using 13–19 weeks old heterozygous male mice (*Bnc2*[+/−]) and their wild-type littermates (*Bnc2*[+/+]) as control.

To induce liver fibrosis, WT mice (females, 10–14 weeks old) were intraperitoneally injected with $CCl_4$ (Sigma-Aldrich) three times a week for 8 weeks. Increasing doses of $CCl_4$ were used according to our previously established procedure[68] as follows: 0.1 mL/kg (week 1), 0.2 mL/kg (week 2), 0.25 mL/kg (week 3), 0.3 mL/kg (week 4) and 0.4 mL/kg (weeks 5–8). The HFSC diet and its associated NASH-like liver phenotype have been described previously[32]. The CDAA-HFSC diet consisted of ad libitum feeding with a choline-deficient, L-amino acid-defined (CDAA) diet supplemented with 35% sucrose, 21% fat, and 2% cholesterol. Control mice were fed a chow diet containing 12% sucrose and 7% fat (Ssniff, custom diets). Mice body weight was measured weekly throughout the course of the experiments.

All animal studies were performed in compliance with EU specifications regarding the use of laboratory animals and have been approved by the Nord-Pas de Calais Ethical Committee (APAFIS#15539-2018053011323354).

### Human liver samples
Fibrotic liver samples were obtained from NASH patients from the ABOS prospective cohort ("Atlas Biologique de l'Obésité Sévère"; ClinGov NCT01129297)[69] and from patients of the TargetOH cohort ("Comparison of Inflammatory Profiles and Regenerative Potential in Alcoholic Liver Disease"; ClinGov NCT03773887; and DC-2008-642)[70,71]. Patients with obesity enrolled in the ABOS cohort were deemed eligible for weight loss surgery[69]. For transcriptomic analyses of biopsies from the ABOS cohorts[36], 53 NASH patients with a Kleiner score F3-F4 were matched for clinical parameters with 53 control patients with obesity and F0 Kleiner score (age, sex, BMI, alcohol, diabetes, statin, fenofibrate; Supplementary Data File 5) as in ref. 72. Follow-up liver biopsies from patients who underwent bariatric surgery were obtained 1 or 5 years after surgery[6]. Liver samples with alcohol-related cirrhosis were obtained from 42 patients with decompensated alcoholic-related cirrhosis who underwent liver transplantation at Huriez Hospital's Liver Unit (Lille, France). Fragments of healthy liver samples were obtained from 20 patients who underwent liver resection for hepatic tumors (control patients; Supplementary Data File 4). The liver samples were immediately fixed for histology or frozen for RNA and protein extraction. Study is authorized by the Lille ethical committee (Lille University Hospital), and informed consent was obtained from all subjects.

### Primary HSC isolation, cell culture, transfection, and treatments
Primary mouse HSCs were purified from C57BL/6J mice (male, 15–18 weeks old) according to the protocol described in[73]. Briefly, livers were digested in situ with 14 mg pronase (Sigma-Aldrich) and 3.7 U collagenase D (Roche) followed by in vitro digestion with 0.5 mg/mL pronase, 0.088 U/mL collagenase D and 0.02 mg/mL DNase I (Roche). HSCs were then separated by a Nycodenz gradient and sorted on a FACS Aria II SORP (BD Biosciences) based on retinoid autofluorescence using the 355 nm laser for excitation and 450/50 nm band-pass filter for detection. Cytometry data were analyzed using FlowJo v10.5.3 (FlowJo, LLC). The purity of HSCs was assessed by measuring the percentage of ultraviolet (UV; retinol autofluorescence) and Desmin-positive cells (Supplementary Fig. 11A).

Primary HSCs and HSC cell lines [human LX2 (Merck, scc064) and murine EMS404 (Kerafast, #EMS404)] were cultured in Dulbecco's Modified Eagle Medium (DMEM) supplemented with 5% fetal bovine serum (FCS, Deutscher). Human primary HSC (Supplementary Data file 10) were grown in DMEM supplemented with 15% FCS. LL-29 cell line was grown into F-12 media supplemented with 15% FCS (Deutscher). AML12 and IHH cells were cultured as described in[74,75]. All cells were grown in the presence of 100 U/mL penicillin-streptomycin (Gibco) at 37 °C in a humidified atmosphere with 5% $CO_2$. As the EMS404 cell line[76] had not been thoroughly used in the literature, we verified the proper expression of MF markers and their ability to respond to TGFβ (Supplementary Fig. 14). For TGFβ stimulation, cells were serum-starved for 24 h in DMEM supplemented with 0.5% FCS and 0.2% BSA before incubation with TGFβ (R&D systems) at 1 ng/mL for 24 h in serum-free media. Primary mouse MF-HSCs were subjected to this protocol 4 days after isolation. Additional treatments with 1 μM verterporfin (Sigma-Aldricht), 0.5 μM MZ1 (Tocris) or 1 μM pomalidomide, thalidomide, CC-885 or Iberdomide (MedChemExpress) were directly added to the growth medium. The TGFβ signaling inhibitors SD208 (Tocris) and SB431542 (Stemgent) were used at 1 and 10 μM, respectively. Spheroids were obtained by growing mouse primary HSCs in U-bottom cell repellent plates (Greiner) for 9 days as described in ref. 47. Non-embedded spheroids were used to ensure inactivation of YAP1[47,77]. Reagents used for cell culture treatments are listed in Supplementary Data File 10.

Transfection of siRNA (used at 20 nM) was performed using lipofectamine 2000 (Invitrogen, Carlsbad, CA, USA) for EMS404 cells, JetPrime (Polyplus transfection) for LX2 cells and HiPerfect (Qiagen) for primary HSCs, according to the manufacturer's instructions. Transfected cells were cultured for 48 h prior to terminal assays. For *Yap1* silencing in mouse primary HSCs, cells were transfected twice (at day 3 and day 6 after isolation) with a siRNA targeting *Yap1* (siYAP1) or non-targeting control siRNA (siCTRL), and cells were harvested 48 h after the last transfection. Control and target siRNAs directed against human or mouse genes were purchased from Thermo Fisher Scientific (Supplementary Data File 10). Plasmid transfection in LX2 cells was performed using JetPEI (Polyplus transfection) according to the manufacturer's instructions. Transfected cells were cultured for 48 h prior to terminal assays. Plasmids used in these experiments are listed in Supplementary Data File 10.

### Isolation of primary mouse liver cell types
Simultaneous purifications of murine primary cell populations from the same livers were performed using a modified version of the method described in ref. 74. Briefly, livers were digested in situ with 100 U/ml of collagenase type IV (Sigma). Hepatocytes (Hep.) were enriched through a 50×*g* centrifugation prior to size-based FACS sorting, whereas the non-parenchymal fraction was further processed for the isolation of hepatic stellate cells (HSC), Kuppffer cells (KC), cholangiocytes (Chol) and endothelial cells (LSEC). The purity of each cell type was assessed by RT-qPCR for cell-specific markers (Supplementary Fig. 11B).

### RNA extraction and RT-qPCR
Tissues were homogenized using a T10 Ultra-Turrax (Ika). The superior right lobe was systematically used for RNA extraction from mouse livers. Total RNA was extracted using Trizol (Eurobio) for cell lines and liver tissues or using the Nucleospin® RNA Plus XS kit (Macherey-Nagel) for primary cells. RNA was reverse-transcribed using the High-Capacity cDNA Reverse Transcription Kit (Applied Biosystem). RT-qPCR was performed in technical triplicates using the SYBR green Brilliant II fast kit (Agilent Technologies) on an Mx3005p apparatus (Agilent Technologies) or a QuantStudio 3 (Applied Biosystems). The specificity of the amplification was checked by recording the dissociation curves, and the efficiency was verified to be above 95% for each primer pair. mRNA levels were normalized to the expression of housekeeping genes as indicated in Supplementary Data File 10, and the fold induction was calculated using the cycle threshold (ΔΔCT) method. The sequences of primers used are listed in Supplementary Data File 10.

### Gene expression microarray
Purified RNA was further digested with rDNase (Macherey-Nagel). RNA integrity and quantity were evaluated using the Agilent 2100 Bioanalyser (Agilent Technologies). RNA was then processed for transcriptomic analysis using Affymetrix GeneChip arrays (HTA 2.0 or MoGene 2.0) according to the manufacturer's instructions. All liquid handling steps were performed by a GeneChip Fluidics Station 450 and GeneChips were scanned with a GeneChip Scanner 3000-7 G using Command Console v4.1.2 (Affymetrix). Quality controls were performed using the Affymetrix expression console.

### Public transcriptomic and functional genomic data recovery
Public data used in this study were downloaded from Gene Expression Omnibus (GEO; http://www.ncbi.nlm.nih.gov/geo/)[78], ENCODE[79], Roadmap epigenome (http://www.roadmapepigenomics.org/)[80], or CistromeDB (http://cistrome.org/db/#/)[41] and are listed in Supplementary Data File 11. RLE normalized expression data (CAGE-seq) from 561 primary cells listed in Supplementary Data File 3 were downloaded from the FANTOM5 website (https://fantom.gsc.riken.jp/5/sstar)[81].

### Transcriptomic data analyses
RNA-seq raw data were processed using a local instance of Galaxy[82]. Data were analyzed using FastQC (http://www.bioinformatics.babraham.ac.uk/projects/fastqc) and reads were trimmed when necessary. Mapping of reads to the human genome (hg38) was achieved using Tophat 2.0.9[83] using the Gencode annotation (GRCh38.84). Read counting was then performed using the Htseq-count tool v1.0.0[84]. Normalization taking into account gene lengths and differential expression analyses were performed using EdgeR v0.0.3[85].

Affymetrix raw data were processed through the GIANT v0.0.2 tools suite[86] on a local instance of Galaxy[82]. Normalization was performed using the apt-probeset-summarize tool of Affymetrix Power Tools (https://www.affymetrix.com/support/developer/powertools/changelog/index.html) using "gc correction+scale intensity+rma". Normalization level was set to "probeset" and probesets assigned to the same gene (NetAffx Annotation Release 36) were averaged for human data while normalization level was set to "Core genes" for mouse data in order to use the "Collapse to gene symbols" function of the GSEA tool. For differential expression analysis, the LIMMA tool from GIANT v0.0.2 was used[87]. When necessary, human gene symbols were attributed to each murine gene using the Orthologue Conversion software (https://biodbnet-abcc.ncifcrf.gov/db/dbOrtho.php).

### Pathway and gene-set enrichment analyses
Enrichments for biological or molecular pathways were defined using Metascape (http://metascape.org)[88] or the ToppGene Suite[89].

Gene-set enrichment analyses (GSEA) were performed using the GSEA software (v3.0) developed at the Broad Institute[50] essentially as described previously[25]. We used 1000 gene-set permutations and the following settings: "weighted" as the enrichment statistic and "difference of classes" or "signal to noise" as the metric for ranking genes. Ranking was performed by the GSEA software using the gene average expression when multiple probesets were present in the microarray. The following gene sets from MSigDB (https://www.gsea-msigdb.org/gsea/msigdb) were used: Biological process v7.2, Reactome v.7.2, and KEGG v7.2. The matrisome gene set comprising 1028 genes encoding for ECM and ECM-associated proteins (or selective subsets including collagen or ECM-regulator encoding genes) was retrieved from http://matrisomeproject.mit.edu/[12]. The human fibrotic liver ECM gene set comprising genes encoding ECM proteins that have been detected by proteomics analyses in fibrotic livers (71 genes) were retrieved from http://matrisomedb.pepchem.org/[51]. In addition to enrichment plots, figures also provide NES and FDR, which are the normalized enrichment score and the false discovery rate provided by GSEA, respectively. The GSEA core enrichment comprising genes that account for the gene set's enrichment signal was retrieved in some of the performed analyses. Random gene lists used in GSEA were obtained using the Random Gene Set Generator tool (https://molbiotools.com/randomgenesetgenerator.php). Similar terms were grouped using an in-house procedure in R[90] or using the GeneSetCluster package[91].

### Gene-module association determination (G-MAD)
Prediction of BNC2 functions was obtained using G-MAD analyses (https://www.systems-genetics.org/gmad)[28], which were performed using default parameters and "*Homo sapiens*" datasets. Additional runs were performed to monitor the G-MAD score obtained for "GO:0044420 extracellular matrix component" in individual organs.

### Single-cell and single-nuclei RNA-seq analyses
Single-cell RNA-seq data of HSCs from control and CCl₄-treated mice have been described in[31]. Briefly, dissociated livers from CCl₄ or vehicle-treated mice were enriched by flow cytometry for LSECs, KCs/MDMs (Kupffer cells/monocyte-derived macrophages), and HSCs prior to 10x Genomics Single-Cell 3' v2 library preparation and

sequencing. Sequencing data were aligned and quantified using the Cell Ranger Single Cell Software Suite (ver. 2.1), and cell types were separated using the Scanpy implementation of the Louvain algorithm. UMAPs were generated using Seurat v. 4.0.3.

Single-nuclei RNA-seq data of liver cells obtained from mice fed a HFSC NASH diet were retrieved from https://www.livercellatlas.org/download.php[34]. Only annotated cells were used, and cell populations with less than 50 individual cells were removed. Counts were then normalized using SCTransform from the Seurat package v. 4.0.3[92]. UMAP were generated using Seurat.

### Protein extraction

Total cell extracts were obtained by washing cells with ice-cold Phosphate-Buffered Saline (PBS) and scraping in Pierce IP lysis buffer (Thermo Fisher Scientific) containing a protease inhibitor cocktail (PIC, Roche). Cell lysates were then sonicated for 5 min (five cycles of 30 s ON/30 s OFF) using Bioruptor (Diagenode), and insoluble material was removed by 10 min centrifugation at 13,000×$g$.

The preparation of protein from the chromatin fraction was performed according to the protocol described in ref. 25. The cytosolic fraction issued from the same samples was also saved and analyzed.

### Protein immunoblotting

Protein concentration was determined using the Pierce[TM] BCA protein assay kit (Thermo scientific), and samples were used for immunoblotting. Regular western blotting or Simple Western immunoassays using the Wes system (Protein Simple), performed as in ref. 25, are described hereafter. Antibodies used and their dilutions are listed in Supplementary Data File 10. Uncropped and unprocessed images are provided in the Source Data file.

**Western blotting.** In total, 40 μg of proteins were separated by 10% SDS-PAGE and immunodetected using the primary antibodies listed in Supplementary Data File 10. Detection was achieved using HRP-conjugated secondary antibodies (Sigma-Aldrich). Images were acquired using the iBright[TM] CL1500 Imaging System (Thermo Fisher Scientific). Quantifications were performed using Image Studio Litev5.2 (LI-COR Biosciences, Lincoln, USA).

**Simple western immunoassays.** Protein extracts (0.4 μg/mL) were run on a Wes system (Protein Simple) according to the manufacturer's instructions. Separation was performed using the 12–230 kDa capillary cartridges. The chemiluminescence-based electrophoretogram was autogenerated, and digitally rendered bands were produced from the chemiluminescent peaks using the Compass software (Protein Simple). Quantifications were obtained using the area under the peak of the protein of interest. Since antibodies against BNC2 had been poorly characterized in the literature, we performed experiments to demonstrate the specificity of #HPA059419 (Sigma) for western blotting (Supplementary Fig. 7).

### Co-immunoprecipitation assays

Co-immunoprecipitation assays were conducted as in[25]. LX2 cells were transfected for 48 h with a plasmid coding for human BNC2 (custom construction, E-Zyvec) and a plasmid coding for constitutively active nuclear YAP1 (YAP1-S127A, Addgene). Alternatively, endogenous proteins were cross-linked by incubating LX2 cells with 1% formaldehyde for 10 min at room temperature. For nuclear extract preparation, cells were rinsed with ice-cold PBS and lysed with hypotonic buffer (20 mM Tris-HCl, pH 8.0, 10 mM NaCl, 3 mM MgCl2, 0.2% NP40) for 5 min at 4 °C. After 5 min centrifugation at 600×$g$, the pellet was resuspended in lysis buffer (25 mM Tris-HCl pH 8, 1 mM EDTA, 1.5 mM MgCl2) and incubated for 30 min at 4 °C. Following 10 min (30 s ON/30 s OFF) sonication with the Bioruptor, insoluble material was removed through 10 min centrifugation at 13000 g. Immunoprecipitation was performed using 2 μg of BNC2 antibody (55220-1-AP from Proteintech or HPA018525 from Sigma-Aldrich) or a control IgG and 500 μg of nuclear extract, which were incubated overnight at 4 °C in dilution buffer (25 mM Tris-HCl pH 8, 1 mM EDTA, 1.5 mM MgCl2). 10 μl of a 1:1 mixture of protein A and protein G magnetic beads (Dynabeads, Thermo Fisher Scientific) blocked overnight in PBS containing 5 mg/mL Bovine serum albumin (BSA) was then added to each sample. After incubating samples 4 h at 4 °C under agitation, beads were washed four times with cold washing buffer (25 mM Tris-HCl, pH 8.0, 150 mM NaCl, 1 mM EDTA, 0.2% NP40). Elution was performed in Laemmli buffer 6× (175 mM Tris-HCl pH 6.8, 15% glycerol, 5% SDS, 300 mM DTT and 0.01% Bromophenol Blue). All buffers were supplemented with PIC (Roche). Input proteins and IPed materials were analyzed by western immunoblotting.

### Chromatin immunoprecipation (ChIP) and ChIP-seq

Chromatin immunoprecipitation was performed as described previously[25]. LX2 cells were fixed for 10 min at room temperature with 1% formaldehyde (Thermo Fisher Scientific) followed by 5 min incubation with 125 mM glycine. After two washes with ice-cold PBS, cell pellets were incubated for 10 min in 0,25% Triton X-100, 10 mM EDTA, 10 mM HEPES and 0.5 mM EGTA followed by 10 min incubation in 0.2 M NaCl, 1 mM EDTA, 10 mM HEPES, 0.5 mM EGTA. Nuclei were then resuspended in Lysis buffer (50 mM Tris-HCl pH 8.0, 10 mM EDTA, 1% SDS) and sonicated for 30 min (three cycles 30 s ON/30 s OFF) using a Bioruptor (Diagenode). All buffers were supplemented with PIC (Roche). Chromatin was diluted tenfold in Dilution Buffer (20 mM Tris-HCl pH 8.0, 1% Triton X-100, 2 mM EDTA, 150 mM NaCl) and incubated overnight at 4 °C with 2 μg of H3K27ac antibody (Active Motif, #39685) or 3 μg of BNC2 antibody (Sigma, HPA018525 or Sigma, HPA059419). The next day, protein A/G sepharose beads (GE Healthcare) were added, and samples were incubated for 4 h at 4 °C under agitation in the presence of 70 μg/mL yeast tRNA (Sigma-Aldrich). Beads were washed three times with RIPA buffer (50 mM HEPES, 1 mM EDTA, 0.7% Na Deoxycholate, 1% NP40 and 500 mM LiCl) containing 10 μg/mL yeast tRNA and once with TE buffer (10 mM Tris-HCl pH 8.0, 1 mM EDTA). DNA was then eluted in 100 mM NaHCO$_3$ containing 1% SDS and incubated overnight at 65 °C for reverse cross-linking. DNA purification was performed using the MinElute PCR purification kit (Qiagen). ChIP and input samples were subjected to high-throughput sequencing and were further analyzed as described hereafter. Selected binding sites were confirmed using qPCR and primers listed in Supplementary Data File 10.

### CoP-seq

The CoP (Column Purified chromatin)-seq procedure was performed essentially as described in ref. 39. In brief, LX2 cells were cross-linked, lyzed and sonicated as described for ChIP except 2% formaldehyde were used. Soluble chromatin was loaded onto MinElute PCR purification kit (Qiagen) and eluates were treated with RNAse A, proteinase K and reverse cross-linked before final DNA purification with the MinElute PCR purification kit. Inputs were processed similarly except for the column-based purification of cross-linked material which was omitted.

### ChIP-seq and CoP-seq data analysis

ChIP-seq and CoP-seq raw data were processed using a local instance of Galaxy[82] essentially as described previously[93]. Data processing involved FastQC analysis (http://www.bioinformatics.babraham.ac.uk/projects/fastqc) and read mapping to hg38 using Bowtie2 version 1.0.0[94]. Genome-wide signal tracks and enriched regions (peak calling) were obtained using model-based analysis of ChIP-seq version 2 (MACS2 v2.1.1.20160309)[95]. Input DNA was used as control, duplicate tags and those mapping to ENCODE blacklisted regions v2[96] were removed and parameters used for peak calling were as described in ref. 93. Signal tracks corresponding to enrichment over input were

obtained using MACS2 bdgcmp. For H3K4me3 ChIP-seq processing, peak calling was performed using the broad option for histone marks ($q < 0.001$). Only H3K4me3 domains which could be assigned to a gene were considered in subsequent analyses, i.e., H3K4me3 domains with predicted gene TSS (GENCODE v24 database) overlapping or within 1 kb. ChIP-seq signals and called peaks were visualized at specific loci using the Integrated Genome Browser (IGB v9.0.1)[97] or were analyzed at large scales using the deepTools v3.3.2[98]. The promoters of housekeeping genes defined in Eisenberg and Levanon, 2013 were used as a control in some of these analyses. The coordinates of their TSS were retrieved using the UCSC Genome Browser[99] and RefSeq as the gene annotation.

The BNC2 cistrome was compared with that of publicly available cistromes contained within the CistromeDB database using the CistromeDB toolkit[41]. « All peaks in each sample » was used for these analyses. De novo motif enrichment analyses were performed using the RSAT peak-motifs tool run with default parameters (http://rsat.sb-roscoff.fr/peak-motifs_form.cgi)[100]. BNC2 target genes were predicted by assigning peaks to genes using GREAT4.0.4[45] or FOCS[49]. With regards to the latest, BNC2 peaks within 2.5 kb of a Gencode v34 TSS[101] were defined as promoter binding sites. The remaining peaks were monitored for predicted promoter interaction within the entire FOCS datasets (http://acgt.cs.tau.ac.il/focs/download.html; enhancers without assigned genes were removed). All retrieved genes were merged defining FOCS predicted BNC2 target genes.

Super-enhancers were defined as in our previous study[102]. First, model-based analysis of ChIP-seq version 2 (MACS2)[95] was used to call peaks in the H3K27ac ChIP-seq data (FDR < 0.15) using input as control. Peaks matching to ENCODE blacklisted regions[96] were discarded. Super-enhancers were then called using retained MACS2-derived peaks and Rank Ordering of Super-Enhancers (stitching set at 12.5 kb)[24,103] leaving out regulatory regions mapping to promoters (TSS +/−2.5 kb) issued from GREAT[45] and using Input DNA as control.

## Bioinformatical identification of MF identity TFs

The cumulative frequency distribution of H3K4me3 domain lengths was used to define the inflexion point of the curve (3017 bp), which served to separate sharp from broad H3K4me3 domains (Fig. 1A)[104]. These broad H3K4me3 domains were assigned to genes using GREAT 2.0[45] (retrieving 1278 genes) in order to only use extremely high-confidence gene predictions as described in[26]. TSSs were used to monitor the distance to the center of the nearest super-enhancer, which were defined using primary MF-HSC H3K27ac ChIP-seq data[18] following a procedure detailed hereabove. Note that we favored the use of broad H3K4me3 over that of super-enhancers to define identity genes to overcome issues highlighted in our previous study including uncertainty about target gene assignment to distal super-enhancers[102].

Next, the length of H3K4me3 domains associated with the 1278 genes in 76 normal human cell types and tissues defined by[26] was retrieved (Supplementary Data File 2). For each gene, the relative H3K4me3 domain length in a given cell type or tissue was obtained by dividing by the mean length of domains associated to this gene in all samples. A heatmap of relative H3K4me3 lengths was generated using the heatmap.2 function of the R package "gplots" (https://cran.r-project.org/web/packages/gplots/) and hierarchical clustering was performed using the hclust function of the R package "Stats" using ward.D2 agglomeration method[105]. This allowed to define three main clusters of genes listed in Supplementary Data File 1. TFs within these clusters were subsequently identified by comparison with the human TFs list provided by the Animal TFDB 2.0 database[106].

Further analysis of TFs from cluster 1 was performed by mining the scientific literature to identify articles linking these TFs to MF or fibrosis using the easyPubMed package (https://www.data-pulse.com/dev_site/easypubmed/) in R. Articles with co-occurrence of a given TF gene name and "fibroblast", "myofibroblast", "fibrosis" or "extracellular matrix" in their title or abstract were retrieved. Results were

visualized on a network generated using Cytoscape[107]. First, the TF network was generated using String 11.0 (https://string-db.org/; "full network", "Medium confidence")[108]. The network was then exported into Cytoscape where TFs without interactions were manually added back and where nodes were colored according to the number of previously retrieved PubMed articles.

## Mouse liver histology and biochemical analyses

**Histology.** Liver slices were fixed with 4% paraformaldehyde for 48 h and embedded in paraffin using a STP 120 Spin Tissue Processor (Microm Microtech). Paraffin-embedded samples were cut at a thickness of 5-μm and sections were transferred on gelatin-coated slides. Fibrosis assessment was carried out by staining liver sections with a 0.1% solution of Sirius red in 1.3% saturated aqueous picric acid solution. To uncertain reliability of Sirius red staining quantifications, we followed guidelines provided in ref. 109. To avoid potential interlobular differences, we systematically used the same lobe (i.e., the right median lobe). Two large sections, including tissue margin and center but excluding the peripheral tissue and vessels, were used per mouse liver. The entire lobe sections were scanned using an Axioscan (Zeiss), and ten fields were randomly chosen (7.5-week experiment) or the entire section (12-week experiment) was used for quantification with Image J version 1.53c software (NIH, https://imagej.nih.gov/ij/). The analysis was systematically done on two different liver sections and all histological analyses were performed blinded.

**Metabolic parameters.** Before sacrifice, blood samples were collected from the retro-orbital sinus of mice. Plasma alanine aminotransferase (ALT) and aspartate aminotransferase (AST) activities were measured using colorimetric assays (Thermo Fisher) on a Konelab 20 (Thermo Fisher).

**Measurement of liver triglycerides.** A weighted piece of liver was homogenized with T10 Ultra-Turrax (Ika) in PBS. Samples were transferred into glass tubes and mixed with a 2:1 chlorofrom:methanol mixture. After centrifugation, upper- and inter-phase were discarded. The lower organic phase was evaporated under nitrogen flow and reconstituted in 1% Triton X-100. Triglyceride content was measured with the LiquiColor Triglycerides Test (Interchim). Protein concentration was measured in parallel using the Pierce™ BCA protein assay kit (Thermo scientific).

## Rapid immunoprecipitation mass spectrometry of endogenous protein (RIME)

LX2 cells were processed using the protocol described in ref. 38. Briefly, $5 \times 10^6$ cells were grown for 2 days before being fixed in 1% formaldehyde (Sigma-Aldrich). Nuclear proteins were extracted in 10 mM Tris-HCl pH 8.0, 100 mM NaCl, 1 mM EDTA, 0,5 mM EGTA, 0,1% Na deoxycholate, 0.5% N-lauroylsarcosine and PIC (Roche). Immunoprecipitation was achieved using Dynabeads (Invitrogen) conjugated with 10 μg of BNC2 antibody (#55220-1-AP, Proteintech or #HPA018525, Sigma-Aldrich) or 10 μg of a non-immune control IgG (#2729, Cell Signaling). Beads were rinsed with RIPA buffer (50 mM HEPES, 1 mM EDTA, 0.7% Na Deoxycholate, 1% NP40 and 500 mM LiCl) and AMBIC solution (Sigma-Aldrich) before trypsin digestion.

Samples were analyzed by coupling a nanoflow liquid chromatography system (nanoElute, Bruker Daltonics) online to a trapped ion mobility spectrometry-quadrupole time of flight mass spectrometer (timsTOF Pro, Bruker Daltonics) equipped with a CaptiveSpray source operating in positive mode. Peptides were loaded on a trapping column (Acclaim PepMap 100, C18, 100 Å, 100 μm × 20 mm, Thermo Scientific) and separated on a reversed-phase C18 column (Aurora, 25 cm × 75 μm i.d., 1.6 μm, IonOpticks). Chromatographic separation was carried out using a gradient of 2-25% of solvent B (0.1% formic acid in acetonitrile) over 90 min, then 37% over 100 min and 95% over

110 min at a constant flow rate of 300 nl/min. The column temperature was controlled a 50 °C. Mass Spectrometry (MS) data were collected over a $m/z$ range of 100–1700, and MS/MS range of 100–1700. LC-MS/MS data were acquired using the PASEF method with a total cycle time of 1.88 s, including 1 TIMS MS scan and 10 PASEF MS/MS scans. Ion mobility coefficient (1/K0) value was set from 0.7 to 1.25 Vs cm$^{-2}$. Singly charged precursors were excluded by their position in the $m/z$-ion mobility plane and precursors that reached a "target value" of 17,000 a.u. were dynamically excluded for 0.2 min. The quadrupole isolation width was set between 2 and 3 $m/z$ depending on precursor $m/z$. The collision energies varied between 20 and 52 eV depending on precursor mass and charge. TIMS, MS operation and PASEF were controlled and synchronized using the control instrument software OtofControl 6.0 (Bruker Daltonik).

Raw MS files were converted into mascot generic files (mgf) and subjected to a Mascot (2.6.2) (Mascot, Matrix Science) search using a target-decoy strategy in order to evaluate the false discovery rate (FDR) of the search. The human database was created with human proteins and present in SwisProt database (created 2019-09-12, containing 20490 target sequences plus the same number of reversed decoy sequences). Search parameters were as follows: trypsin was set as the cleavage enzyme and a maximum of one miscleavage was allowed. Cysteine carbamidomethylation was set as a fixed modification, whereas methionine oxidation, was set as a variable modification. The peptide mass tolerance (tolerance of mass measurement for precursor ion) was set to 15 ppm and the MS/MS mass tolerance (tolerance of mass measurement for fragment ion) set to 0.05 Da. Proline pipeline (http://www.profiproteomics.fr/proline/) was used to validate the identification results. This statistical validation was performed using the Target Decoy approach, which consists in creating a "decoy" sequence that does not exist in nature. The false identifications thus distribute evenly between the real database (target) and the decoy one, and so the number of decoy hits can be used to estimate the false discovery rate (FDR = decoy hits/decoy hits + target hits) and eliminate false positives. Peptide and protein identification validation parameters were set as follows: minimal length of seven amino acid, score ≥20, pretty rank ≤1, and FDR ≤ 1.

Only proteins detected in the BNC2 RIME, at least two out of three biological replicates, but in none of the non-immune control IgG RIME experiments were considered. Common non-specifically detected proteins in affinity purification mass spectrometry data from the Contaminant Repository for Affinity Purification (human CRAPome2.0) database[110] were also discarded. BNC2 interactors were further filtered using the list of human transcriptional regulators from the AnimalTFDB3.0 database[111]. BNC2 interactors were clustered based on percent protein coverage in RIME assays using the heatmap.2 function of the R package "gplots" (https://cran.r-project.org/web/packages/gplots/) as described hereabove for H3K4me3 domain lengths. Biological triplicates were obtained using the anti-BNC2 antibody 55220-1-AP (Proteintech) and an additional experiment was performed using the anti-BNC2 antibody HPA018525 (Sigma-Aldrich). All non-filtered detected proteins for individual samples are provided in Supplementary Data Files 6 and 8.

## RNA in situ hybridization

RNAscope assays on paraffin-embedded liver sections were performed using the RNAscope® Multiplex Fluorescent Reagent Kit v2 according to the manufacturer's instructions (Bio-Techne). Briefly, the paraffin-embedded tissue sections were deparaffinized and pre-treated with hydrogen peroxide. Sections were incubated in RNAscope® 1× Target Retrieval Reagent for 15 min at 99 °C and treated with RNAscope® Protease Plus for 30 min at 40 °C in the HybEZ oven sequentially. Samples were then incubated with RNAscope® Probe-Hs-BNC2-C1 (#496801) and Probe-Hs_COL1A1-C2 (#401891-C2) for 2 h at 40 °C. After three steps of amplification, RNAscope® HRP-C1 was added on the sections that were incubated with Opal 650 (Akoya Biosciences) for

30 min. HRP-C2 was then added followed by incubation with Opal 570. When specified, RNA in situ hybridization was followed by immunofluorescence staining using an anti-ACTA2 antibody (ab76126, Abcam:1:50). Alcohol-related liver cirrhosis samples were analyzed using a CSU-W1 Spinning Disk (Gataca) with a 60x CFI PLAN APO LBDA objective (resolution 0.110 μM) and images were analyzed using ImageJ1.53c (https://imagej.nih.gov/ij/). For liver biopsies from NASH patients before and after bariatric surgery, the entire liver sections were scanned using the Axioscan (Zeiss) and images were analyzed using the ZEN 2 software (Zeiss). Quantification of red and white dots; illustrating the number of *COL1A1* and *BNC2* transcripts, respectively, was performed using ImageJ1.53c (https://imagej.nih.gov/ij/). The number of cells in each section was obtained by counting DAPI-positive nuclei.

## Structural model of the BNC2-CRBN interaction and docking analyses

A structural model of the interaction between BNC2 zinc-finger 1 (BNC2ZF1) and CRBN was built using the crystallographic structure of DDB1-CRBN-pomalidomide bound to IKZF1 (Protein Data Bank ID 6h0f)[52] and used for docking of thalidomide and its derivatives. First, we chose an experimentally resolved zinc finger-CRBN complex on the basis of its resolution and co-crystallized ligand, settling for DDB1-CRBN-pomalidomide bound to IKZF1 (Protein Data Bank ID 6h0f)[52]. Out of the six BNC2 zinc fingers, four were below a 25% identity threshold with the co-crystallized IKZF1 zinc finger. The two remaining BNC2 zinc fingers were the first (29% identity, 38.7% similarity) and fifth (25.8% identity and 38.7% similarity). We opted to keep the first (BNC2ZF1), as it was closer to the reference. BNC2ZF1 was modeled on the RMN structure of human zinc fingers and homeoboxes 1 (Protein Data Bank ID 2ghf; 20.83% identity and 50% similarity with BNC2ZF1). Interestingly, even with a higher identity percentage than the template used to build BNC2ZNF1, the model resulting from a homology with IKZF1 was marginally less interesting, with a slightly higher overall deviation from the Ramachandran optimal parameters. The BNCZF1-CRBN complex was next built by superimposing the BNC2 and IKZF1 zinc fingers and exchanging the co-crystallized IKZF1 for BNCZF1 and adjusting the side chains by a quick geometry optimization in two steps after adding hydrogens and partial charges. First, the side chains were adjusted by 1000 steps of steepest descent, then the whole zinc finger and the side chains of the DDB1-CRBN-pomalidomide complex were subjected to another 1000 steps of steepest descent. This protocol avoided a deformation of the skeletons of the proteins while minimizing the clashes between side chains. The resulting complex was used as the target for a docking study with GOLD (Genetic Optimisation for Ligand Docking)[112], with the binding site defined as 10 Å sphere around the co-crystallized pomalidomide. A set of ligands was docked: pomalidomide as a control, the shorter thalidomide, iberdomide which was similar in size to CC-885, and CC-885. In all, 30 poses were generated for each compound. The docking was assessed on the basis of the number of clusters of closely related poses, the score of the clusters, and their spread around their average pose. Apart from CC-885, all the compounds were built in the same configuration as the co-crystallized pomalidomide (S enantiomer).

## Statistical analyses

Statistical analyses were performed using the Prism software (GraphPad) and R[105]. The specific tests and corrections for multiple testing that were used are indicated in the figure legends. Unless specified in the figure legends, statistical significance was displayed as follow *$P < 0.05$, **$P < 0.01$, and ***$P < 0.001$.

## Reporting summary

Further information on research design is available in the Nature Research Reporting Summary linked to this article.

## Data availability

Transcriptomic and cistromic data generated in this study have been deposited into Gene Expression Omnibus under SuperSeries accession number GSE185529. The mass spectrometry proteomics data generated in this study have been deposited to the ProteomeXchange Consortium via the PRIDE [1] partner repository with the dataset identifier PXD003624. Datasets analyzed in this study (Supplementary Data File 11) are available through the Gene Expression Omnibus (GSE68108[18], GSE58680[113], GSE38103[43], GSE61852[44], GSE68108[18], GSE111059[114], GSE63626[115], GSE145086[31], GSE192742[34]), ENCODE (https://www.encodeproject.org/experiments/ENCSR507UDH/)[79], FAN-TOM5 (https://fantom.gsc.riken.jp/5/sstar)[81], the matrisome database (http://matrisomeproject.mit.edu/ and http://matrisomedb.pepchem.org/)[51], and manuscript supplementary data (https://www.nature.com/articles/ng.3385 and https://www.atsjournals.org/doi/10.1164/rccm.201712-2410OC)[26,116]. Source data are provided with this paper.

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

## Acknowledgements

The authors are indebted to Julien Devassine and Kim Letten of the EOPS2 animal facility (Plateformes Lilloises en Biologie et Santé (PLBS) - UMS 2014 - US 41, Univ. Lille), Sophie Salomé and Nathalie Jouy of the BioImaging Center Lille (BICeL) platform (Plateformes Lilloises en Biologie et Santé (PLBS) - UMS 2014 - US 41, Univ. Lille) for their assistance. The authors also would like to thank members of Inserm U1011 including Drs M. Johanns, L. L'Homme, N. Hennuyer, X. Maréchal, M. Rosa and M. Gimenez as well as F. Maggiotto (INFINITE, Lille) for technical help or helpful discussions. This work was supported by the Agence Nationale de la Recherche (ANR) grants "HSCreg" (ANR-21-CE14-0032), "MEdicAL" (ANR-21-CE17-0016), "DeCodeNASH" (ANR-20-CE14-0034), "PreciNASH" (ANR-16-RHUS-0006-PreciNASH), French Proteomic Infrastructure (ProFI; ANR-10-INBS-08–03) and "European Genomic Institute for Diabetes" E.G.I.D. (ANR-10-LABX-0046), a French State fund managed by ANR under the frame program Investissements d'Avenir I-SITE ULNE/ANR-16-IDEX-0004 ULNE. This project was also co-funded by The European Union under the European Regional Development Fund (ERDF) and by the Hauts de France Regional Council (contract 20000007 and contract 17003781), the MEL (contract_2020_ESR_02 and contract_2016_ESR_05) and the French State (contract no. 2019-R3-CTRL_IPL_Phase3 and contract no. 2017-R3-CTRL-Phase 1). Work in INFINITE was supported by a grant from the Fondation Recherche Médicale (FRM) (Equipe labellisée, EQU202003010299). BS is a recipient of an Advanced ERC Grant (694717).

## Author contributions

Conceptualization: M.B.-G., F.P.Z., F.P., L.D., B.S., P.L., and J.E.; methodology: M.B.-G., C.B., F.P.Z., J.D.-C., C.G., M.B.S., G.L., J.-M.S., A.F., M.P., J.V., L.F., L.C.N.-W., B.P., and J.T.H.; software: G.L., J.D.-C., J.V., and K.R.; validation: M.B.-G., C.B., J.D.-C., and C.G.; formal analysis: M.B.-G., J.D.-C., and J.E.; investigation: M.B.-G., C.B., F.P.Z., J.D.-C., C.G., M.B.S., J.-M.S., A.F., M.P., J.V., N.V., E.W., A.D., E.B., A.K.C., C.G., E.V., R.P., K.R., and J.T.H.; resources: L.G., P.M., V.R., R.C., S.C., F.P., L.D., V.G., and E.L.; data curation: M.B.-G., J.D.-C., and J.E.; writing—review & editing: M.B.-G. and J.E. with inputs from co-authors; visualization: G.L., M.B.-G., C.B., F.P.Z., A.F., K.R., J.C.-D., and J.E.; project administration: B.S., P.L., and J.E.; funding acquisition: B.S., P.L., and J.E.

## Competing interests

The authors declare no competing interests.

## Additional information

**Marie Bobowski-Gerard** [1], **Clémence Boulet**[1], **Francesco P. Zummo**[1], **Julie Dubois-Chevalier**[1], **Céline Gheeraert**[1], **Mohamed Bou Saleh**[2], **Jean-Marc Strub**[3], **Amaury Farce** [2], **Maheul Ploton** [1], **Loïc Guille** [1], **Jimmy Vandel**[1], **Antonino Bongiovanni** [4], **Ninon Very** [1], **Eloïse Woitrain**[1], **Audrey Deprince**[1], **Fanny Lalloyer**[1], **Eric Bauge**[1], **Lise Ferri**[1], **Line-Carolle Ntandja-Wandji**[2], **Alexia K. Cotte**[1], **Corinne Grangette**[5], **Emmanuelle Vallez**[1], **Sarah Cianférani** [3], **Violeta Raverdy**[6], **Robert Caiazzo**[6], **Viviane Gnemmi**[7], **Emmanuelle Leteurtre**[7], **Benoit Pourcet**[1], **Réjane Paumelle**[1], **Kim Ravnskjaer** [8,9], **Guillaume Lassailly**[2], **Joel T. Haas** [1], **Philippe Mathurin**[2], **François Pattou** [6], **Laurent Dubuquoy** [2], **Bart Staels** [1], **Philippe Lefebvre**[1,10] & **Jérôme Eeckhoute** [1,10] ✉

[1]Univ. Lille, Inserm, CHU Lille, Institut Pasteur de Lille, U1011-EGID, F-59000 Lille, France. [2]Univ. Lille, Inserm, CHU Lille, U1286 - INFINITE - Institute for Translational Research in Inflammation, F-59000 Lille, France. [3]Laboratoire de Spectrométrie de Masse BioOrganique, CNRS UMR7178, Univ Strasbourg, IPHC, Infrastructure Nationale de Protéomique ProFI - FR2048, 67087 Strasbourg, France. [4]Univ. Lille, CNRS, Inserm, CHU Lille, Institut Pasteur de Lille, US 41 - UAR 2014 - PLBS, F-59000 Lille, France. [5]U1019-UMR 9017-CIIL-Centre d'Infection et d'Immunité de Lille, Institut Pasteur de Lille, Université de Lille, CNRS, Inserm, CHU Lille, F-59000 Lille, France. [6]Univ. Lille, Inserm, CHU Lille, Institut Pasteur de Lille, U1190-EGID, Translational Research in Diabetes, Lille, France. [7]Service d'anatomopathologie, Centre Hospitalier Universitaire de Lille, Université de Lille, INSERM UMR-S 1172, Lille, France. [8]Department of Biochemistry and Molecular Biology, University of Southern Denmark, Odense 5230, Denmark. [9]Center for Functional Genomics and Tissue Plasticity (ATLAS), University of Southern Denmark, Odense 5230, Denmark. [10]These authors jointly supervised this work: Philippe Lefebvre, Jérôme Eeckhoute. ✉e-mail: jerome.eeckhoute@inserm.fr

