## [Peer Review File · Nature Communications]

Title: Functional genomics uncovers the transcription factor BNC2 as a cornerstone of myofibroblastic activation in fibrosisREVIEWER COMMENTS

Reviewer #1 (Remarks to the Author):

This is an interesting study identifying Bnc2 as a novel transcription factor responsible for the activation of hepatic stellate cells and liver fibrosis. Although the authors analyzed the role of Bnc2 in Bnc2-heterozygous mice, the impact of Bnc2 heterozygous loss is minor and additional studies using primary hepatic stellate cells will strengthen the conclusion of this study. In addition, the authors did not analyze or discuss the possible role of Bnc2 in ECM degradation pathways.

1. In Fig. 3, the authors showed recruitment of BNC2 to the SMAD3/YAP1 binding sites in COL1A1 and BNC2 genes. Is BNC2 also recruited to the promoter of ACTA2 gene, which is a marker for the activation of hepatic stellate cells?
2. The authors analyzed matrisome genes transcriptionally regulated by BNC2. During activation, hepatic stellate cells also change expression of TIMP1 and MMPs. Did the authors find transcriptional regulation of these genes involved in matrix degradation pathways by BNC2? Additional data for these genes will be desirable.
3. Fig. 3K, does a TGFb receptor inhibitor suppress the Bnc2 expression?
4. Since Bnc2-null mice are lethal, the authors used Bnc2-heterozygous and wild type mice to examine the role of Bnc2 in liver fibrosis in Fig. 5. The fibrotic area estimated by Sirius red staining shows minor reduction. Hydroxy proline assay will be necessary to validate this finding.
5. Primary mouse hepatic stellate cells spontaneously differentiate into myofibroblasts in culture. Does suppression of Bnc5 by siRNA reduce their activation in vitro? Or do primary hepatic stellate cells from Bnc5-heterozygous mouse livers show less fibrosis in culture? Since CC-885 does not work in mice, validation by siRNA or gene-mutated mice will be necessary to warrant the in vivo data.

Reviewer #2 (Remarks to the Author):

Comments related to mass spectrometry-guided findings:

1. The authors need to provide raw data of mass spectrometry of RIME experiments.
2. For the RIME experiments, have the authors used n>3 biological independent replicates for each group? Statistical details are lacking. The methods section related to RIME experiments is also too simple for the reviewer to judge the soundness of the experimental design and the results.
3. Considering the differences in genetic regulation mechanisms among species, the reviewer suggests

the authors perform RIME experiments for murine-derived cell line or mouse primary MFs as well.

4. In supplementary table 6, the authors should provide more detailed information, such as protein-protein interaction sites, which I believe cannot be collected via RIME, abundance of proteins etc.

Comments related to biological studies:

1. The authors found that the lower level of YAP1 and Bnc2 in 3D-spheroids of primary murine Q-HSCs when compared to 2D-cultures, their results are consistent with publication by Mannaerts, I. et al (2015). However, Thomas B. et al (2019) paper concluded that both YAP1 and Bnc2 level and the stiffness increased in 3D-spheroids of SCCs. In order to address the conflicted findings, it would be nice to discuss the results (Line 242-244).

2. The author used heterozygous Bnc2^{+/-} as animal model to study BNC2 controls the development of liver fibrosis. Since HSCs-specific gene knock-out technology is well established, it would be nice to confirm the results on HSCs-specific BNC2 knock-out mice (Line 299-301).

Reviewer #3 (Remarks to the Author):

The authors suggest a transcription factor, BNC2, as a novel transcriptional regulator in myofibroblastic activation in fibrosis. They combined computational analysis using public data as well as in-house genome-wide data to systematically assess and prioritize potential key players in fibrosis followed by in vitro/in vivo experimental validation. It was quite fun overall to read the manuscript and to see the incremental build-up of the story about Bnc2. The manuscript is well written, the evidence is convincing, and the conclusion has novelty and impact. A few methodological clarifications and additional mining of the genomewide data would make it publishable.

1. The authors used the notion of broad vs. sharp peaks of H3K4me3 to identify so-called “identity genes.” Although H3K4me3 is a representative promoter marker, H3K27ac is also enriched at promoter as well as enhancers. And superenhancers (SE) defined by H3K27ac are also known to represent the “key genes” of cell types. It is conceivable that SE analysis would capture Bnc2 as well given the similarity of the H3K4me3 profiles and H3K27ac at the key gene locus, but FigS1B shows that H3K27ac displays both higher and broader enrichment. Can you perform SE analysis using H3K27ac and compare the associated genes with the currently identified genes by H3K4me3? If H3K27ac also captures a similar set of key genes, that would strengthen the gene prioritization; if not, that would also suggest distinct epigenetic roles in fibrosis, which would be also interesting on its own.

2. In Fig S8B motif analysis:

a. Please include more detailed statistics including Hit%

b. Any functional implications from the motif contents other than the similarity with previous publications??

3. In Fig 3D, did the author test SMAD as well for co-IP?

4. In Fig 4A&b:

a. What data are these? RNA-seq? or microarray? Method section contains very routine procedures only. Please clarify.

b. What about down/up genes defined by FC or FDR in a canonical way?

c. What are the overall characteristics of siBNC2-induced genes?

5. In line 251: What does “deregulated” means? Does it imply both down and up-regulation?

6. In Fig 4A:

a. How exactly were these terms retrieved? Are these the top significant gene sets? Or fibrosis-related terms only?

b. NES values are > 1 . Does this mean that they are induced in siBNC2?

7. In Fig 4B:

a. How exactly is this different from Fig 4A except that it is “molecular function.” For any gene list, we can assess both the biological process and molecular function. Why only one in Fig 4A and the other in Fig 4B?

b. Add statistics like Fig 4A if applicable.

8. In Fig 4I: One legend is missing for light green.

We would like to thank the reviewers for their insightful comments. We provide below a point-by-point response to the reviewers' criticisms and suggestions for improvement. We also provide a marked version of the revised manuscript where main changes are highlighted in red.

Reviewer #1 (Remarks to the Author):

This is an interesting study identifying Bnc2 as a novel transcription factor responsible for the activation of hepatic stellate cells and liver fibrosis. Although the authors analyzed the role of Bnc2 in Bnc2-heterozygous mice, the impact of Bnc2 heterozygous loss is minor and additional studies using primary hepatic stellate cells will strengthen the conclusion of this study. In addition, the authors did not analyze or discuss the possible role of Bnc2 in ECM degradation pathways.

1. In Fig. 3, the authors showed recruitment of BNC2 to the SMAD3/YAP1 binding sites in COL1A1 and BNC2 genes. Is BNC2 also recruited to the promoter of ACTA2 gene, which is a marker for the activation of hepatic stellate cells?

We did not find evidence of BNC2 recruitment to the promoter of the *ACTA2* gene. Please see below a comparison of the BNC2, YAP1 and SMAD3 ChIP-seq signals together with those of H3K27ac and H3K4me3 at the *COL1A1* and *ACTA2* genes. Surprisingly and opposite to *COL1A1*, *ACTA2* lacks widespread and strong H3K27ac labelling and recruitment of the myofibroblast transcriptional regulators. In particular, BNC2 recruitment was not evidenced at the *ACTA2* gene regulatory regions (highlighted by the dotted boxes). Further pointing to *ACTA2* not being directly regulated by BNC2, its expression was not reduced in the transcriptomic data issued from LX2 cells transfected with siBNC2. Recent single-cell transcriptomic studies of both mouse and human MFs have defined that *ACTA2* expression does not strongly associate with that of *Col1a1* including fibroblast subsets with low *ACTA2* despite high expression levels of collagen genes (Tsukui et al. *Nat Commun* 2020, Pakshir et al. *J Cell Sci.* 2020, McAndrews et al. *EMBO J* 2022). Moreover, treatment of HSCs with drugs inactivating different pro-fibrotic signaling pathways showed variable and sometimes antagonistic effects on *Acta2* and *Col1a1* expression (Sakai and Yoshimura, *Biol Pharm Bull* 2020). Hence, despite being a well-accepted marker of MFs, including activated hepatic stellate cells (HSCs), *ACTA2* transcriptional regulation appears to be controlled by different means when compared to that of *COL1A1*.

2. The authors analyzed matrixome genes transcriptionally regulated by BNC2. During activation, hepatic stellate cells also change expression of TIMP1 and MMPs. Did the authors find transcriptional regulation of these genes involved in matrix degradation pathways by BNC2? Additional data for these genes will be desirable.

The reviewer is correct in pointing out that myofibroblasts control the ECM both through production of ECM constituents and expression of ECM remodeling/degrading enzymes. In order to investigate if BNC2 controls the expression of these 2 different categories of ECM-related genes, we performed additional mining of the transcriptome changes induced by BNC2 silencing in both LX2 (from the original submission) and EMS404 (novel data acquired during revision of the manuscript; see Fig.4E-F and response to reviewer 2's point 3) cells. We found that BNC2 silencing induced a significant

decrease in expression of both genes encoding collagens, main ECM constituents, and of genes encoding ECM-remodeling enzymes (Supplementary Fig.15). In addition to LOXL1, which we further analyzed by RT-qPCR (Fig.4G-H), several MMPs/TIMPs (MMP11, MMP16, TIMP4) were also decreased. This is now indicated in the revised manuscript lines 278-280.

3. Fig. 3K, does a TGF β receptor inhibitor suppress the *Bnc2* expression?

To answer this question, we used two different TGF β receptor kinase inhibitors, i.e. SB431542 and SD208 (Haque and Morris, *Hum Vaccin Immunother* 2017). Treatment for 24h with these compounds led to a significant decrease in expression of *Bnc2* as shown in Fig.3M. As expected, these inhibitors therefore trigger a mirror effect when compared to induction of *Bnc2* expression by treatment with TGF β (Fig.3L). These novel data are described lines 248-249.

4. Since *Bnc2*-null mice are lethal, the authors used *Bnc2*-heterozygous and wild type mice to examine the role of *Bnc2* in liver fibrosis in Fig. 5. The fibrotic area estimated by Sirius red staining shows minor reduction. Hydroxy proline assay will be necessary to validate this finding.

As detailed hereafter, in order to further ascertain that *Bnc2* heterozygosity leads to reduced collagen deposition, we provide the results of a novel mouse experiment in the revised manuscript. In our study, we have not used quantification of hydroxyproline, a collagen post-translational modification whose levels may be influenced by differential activities of prolyl hydroxylases in addition to the amount of collagen *per se*. Indeed, hydroxylation of proline is regulated by co-factors of iron, oxygen and ascorbic acid, with dependency on the cellular metabolic status and oxidative stress (Rappu P et al. *Essays in Biochemistry* 2019). As a consequence, pathophysiological differences in qualitative and quantitative regulation of collagen hydroxylation have been reported (Rappu P et al. *Essays in Biochemistry* 2019; Tang et al. *Clin Cancer Res* 2018; Clift et al. *Scientific Rep* 2021; Kirchner et al. *PLoS One* 2021). In this context, we favored the use of fibrillar collagen detection *in situ* using Sirius red staining. We would like to take this opportunity to state actions we had undertaken to ascertain the reliability of our Sirius red staining quantifications which were missing in our initial description of this procedure. Indeed, we made sure to follow guidelines provided in Clapper JR et al. *Am J Physiol Gastrointest Liver Physiol* 2013. Hence, to avoid potential interlobular differences, we systematically used the same lobe (i.e. the right median lobe). Two large sections including tissue margin and center but excluding the peripheral tissue and vessels were used per mouse liver and entire sections were used for quantification. Finally, all Sirius red quantifications were performed blinded. The Materials and Methods section of the manuscript has been amended to include this more thorough description of fibrillar collagen quantification by Sirius red staining (lines 611-618).

In order to further define that *Bnc2* controls collagen deposition in the liver of mice fed the HFSC pro-fibrotic diet, we have performed an entirely new experiment whose results are displayed in Supplementary Fig.19 and referred to in the revised manuscript lines 318-320. When compared to our initial experiment shown in Fig.5, we have extended the HFSC feeding period (12 weeks instead of the 7.5 weeks used in our initial experiment) in order to promote greater collagen deposition and we have further increased the number of animals involved (15-19 mice per group). Using Sirius red staining in order to specifically quantify fibrillar collagen, we replicated the finding that *Bnc2* heterozygosity leads to reduced liver collagen deposition (Supplementary Fig.19B). In line with reduced collagen deposition, impaired HSC molecular activation was also further confirmed in the livers of *Bnc2*^{+/-} mice. Indeed, RT-qPCR analyses again showed lower expression of matrix genes in the livers of *Bnc2*^{+/-} mice (Supplementary Fig.19C). This finding is reminiscent of our previously obtained gene expression data (including transcriptomic analyses; Fig.5E-H). Even though the use of *Bnc2* heterozygote mice intrinsically leads to moderate consequences on HSC activation and fibrosis development, we feel that our combined phenotypic and molecular characterization of the liver of mice from 2 entirely independent experiments (systematically involving >10 mice per group) unequivocally demonstrate a role for *Bnc2* in HSC-mediated fibrillar collagen deposition.

5. Primary mouse hepatic stellate cells spontaneously differentiate into myofibroblasts in culture. Does suppression of Bnc5 by siRNA reduce their activation in vitro? Or do primary hepatic stellate cells from Bnc5-heterozygous mouse livers show less fibrosis in culture? Since CC-885 does not work in mice, validation by siRNA or gene-mutated mice will be necessary to warrant the in vivo data.

We agree with the reviewer that defining a role for BNC2 specifically in primary hepatic stellate cells (HSCs) is needed in addition to the *in-vivo* data. Our manuscript presents the results of BNC2 silencing experiments both in primary mouse and human HSCs (mouse HSCs from 3 independent preparations and human HSCs from 3 different donors were used). Reminiscent of what we observed in HSC cell-lines and in line with the *in vivo* data, we found that BNC2 controls matrisome gene expression in primary HSCs (Fig.4G and H). The labels in Fig.4 have been revised to more clearly state that both mouse and human cells, including primary cells, were used.

Reviewer #2 (Remarks to the Author):

Comments related to mass spectrometry-guided findings:

1. The authors need to provide raw data of mass spectrometry of RIME experiments.

As indicated in the “Data availability” section, all raw mass-spectrometry data have been made publicly available through the ProteomeXchange database. All identified proteins are provided in Supplementary Tables 6 and 7. Reviewer account details for PRIDE data: Username: reviewer_pxd003624@ebi.ac.uk; Password: OkBneJaO.

2. For the RIME experiments, have the authors used n>3 biological independent replicates for each group? Statistical details are lacking. The methods section related to RIME experiments is also too simple for the reviewer to judge the soundness of the experimental design and the results.

As suggested by the reviewer, the revised manuscript includes a more thorough description of the RIME experiments and of obtained data using biological replicates. Indeed, we now provide a novel panel in Fig.3 (Fig.3D), which shows the result of RIME data obtained using 2 different BNC2 antibodies, i.e. biological triplicate data using antibody 55220-1-AP (Proteintech) and an additional experiment performed using antibody HPA018525 (Sigma-Aldrich). As now stated in the revised manuscript (lines 632-643), only transcriptional regulators detected in the BNC2 RIME, at least 2 out of 3 biological replicates obtained with antibody 55220-1-AP (Proteintech), but in none of the non-immune control IgG RIME experiments were considered. Common non-specifically detected proteins in affinity purification mass spectrometry (CRAPome2.0; Mellacheruvu D et al. *Nature Methods* 2013) were also discarded. Overall, the data point to cofactors involved in signal transduction from stiff ECM including the Hippo/YAP1 pathway as defined through novel gene enrichment analyses shown in Supplementary Fig.10. More specifically, the cofactor YAP1 was detected in all but 1 BNC2 RIME samples and was recovered with the 2 different BNC2 antibodies used in these assays. Together with the co-immunoprecipitation assays shown in Fig.3E and Supplementary Fig.9D (performed using the same 2 different BNC2 antibodies mentioned hereabove), our data argue that YAP1 is a *bone fide* novel BNC2 interactor.

3. Considering the differences in genetic regulation mechanisms among species, the reviewer suggests the authors perform RIME experiments for murine-derived cell line or mouse primary MFs as well.

Our initial attempts to perform BNC2 RIME and ChIP-seq were performed in parallel in human LX2 and mouse EMS404 MF-HSC cells. We found that the BNC2 antibodies were less efficient at immunoprecipitating mouse BNC2 when compared to human BNC2 preventing us from obtaining robust data in the EMS404 cells. Moreover, in our experience, the limited yield of mouse primary HSCs does not allow to retrieve sufficient materials for successful RIME experiments.

However, our study has leveraged a compendium of omics data obtained in samples from both mouse and human origins and extending to MFs beyond HSCs. The fibrogenic signaling pathways which we functionally link to BNC2, i.e. the TGFbeta and Hippo/YAP1 pathways, are well-established to operate both in mouse and human myofibroblasts (Pakshir et al. *J Cell Sci* 2020). For instance, data

presented in Fig.3H-M, which functionally define *Bnc2* as a downstream target of TGFbeta and YAP1, were obtained using mouse HSCs (this has been clarified in the figure legends). In order to further establish that BNC2 controls conserved biological pathways in mouse and human cells, we have monitored how BNC2 silencing modulates the transcriptome of the mouse EMS404 cells. Importantly, reminiscent of our findings in the human LX2 cells, we found that BNC2 silencing in mouse EMS404 cells leads to down-regulation of matrisome genes including both ECM constituents and ECM modulators (Revised manuscript lines 275-280; novel panels in Fig.4E-F and Supplementary Fig.15). Therefore, despite undisputable potential species-specific regulation of MF activities, we feel our study establishes a functional connection between BNC2 and canonical pro-fibrotic signaling pathways and target genes operating both in mouse and human MFs. The novel transcriptomic data have been added to SuperSeries GSE185529, which gather all transcriptomic and cistromic data related to our manuscript.

4. In supplementary table 6, the authors should provide more detailed information, such as protein-protein interaction sites, which I believe cannot be collected via RIME, abundance of proteins etc.

Details about detected proteins in each individual RIME sample are now provided in Supplementary Tables 6 and 7. For each protein, included information comprise the percent coverage as well as the number of both total and unique peptides detected. Defining protein-protein interaction sites is indeed beyond the reach of RIME and requires approaches where cross-linked peptides are specifically enriched (Richards AL et al. *Mol Syst Biol* 2021).

Comments related to biological studies:

1. The authors found that the lower level of YAP1 and *Bnc2* in 3D-spheroids of primary murine Q-HSCs when compared to 2D-cultures, their results are consisted with publication by Mannaerts, I. et al (2015). However, Thomas B. et al (2019) paper concluded that both YAP1 and *Bnc2* level and the stiffness increased in 3D-spheroids of SCCs. In order to address the conflicted findings, it would be nice to discuss the results (Line 242-244).

In our study, we used non-embedded spheroids which were obtained by growing cells in U-bottom cell repellent plates. This is different from many studies in the cancer field, including the work from Pankova et al. *EMBO J* 2019 (which we believe is the reference the reviewer had in mind), where embedded spheroids are used in order to mimic the tumor micro-environment. Embedded spheroids are often used to study tumor cell invasion and are therefore often referred to as the “spheroid invasion assay” (Tevis K.M. et al. *Adv Biosyst.* 2017). A fundamental difference between the non-embedded and embedded approaches is that the latest provides a scaffold material which will influence cell characteristics at least in part due to mechanical reinforcement. Pankova et al. *EMBO J* 2019 report YAP nuclear staining in H1299 cells grown as spheroids embedded into Collagen Type I, which has been shown to be sufficient to drive YAP re-activation in mesenchymal cells initially grown as non-embedded spheroids (Komatsu et al. *Stem Cell Research & Therapy* 2018). Hence, the use of Collagen Type I embedded in Pankova et al. (*EMBO J* 2019) versus non-embedded spheroids in our work most probably account for the differences in YAP activation. We have clarified in the Results section of our revised manuscript that we used non-embedded spheroids (line 245) and indicated in the Methods section that this was used to inactivate YAP1 based on Mannaerts I. et al *J Hepatol* 2015 and Pankova et al. *EMBO J* 2019 (line 446).

2. The author used heterozygous *Bnc2*^{+/-} as animal model to study BNC2 controls the development of liver fibrosis. Since HSCs-specific gene knock-out technology is well established, it would be nice to confirm the results on HSCs-specific BNC2 knock-out mice (Line 299-301).

The most common way to inactivate a gene in HSCs involves crossing floxed mice for the gene of interest with mice expressing the CRE recombinase under the control of promoters (*Pdgfrb* or *Lrat* promoters) active in HSCs. Floxed mice allowing to delete the *Bnc2* gene have not been established yet and no purely HSC-specific promoter construct driving CRE expression are available (*Pdgfrb*-Cre or *Lrat*-Cre mice express CRE in non-liver cells; Ravnskjaer K, unpublished observations). In this context, we

used mice heterozygous for *Bnc2* in our study. Importantly, we ascribed the effect of *Bnc2* heterozygosity on liver fibrosis to its role in MF-HSCs since we showed that *Bnc2* expression is specifically induced in HSCs within fibrotic mouse livers (Fig.2B and Supplementary Fig.5A-D). This has been further validated using the re-analysis of single-nuclei transcriptomic data obtained from the livers of a NASH mouse model recently published in Williams et al. *Cell* 2022 (Supplementary Fig.6). Moreover, heterozygous mice may be less prone to adaptation/compensation with regards to homozygous deletion of both *Bnc2* alleles and this strategy is relevant for transcription factors since these are the genes which are the most sensitive to gene dosage (Ni Z et al. *Frontiers in Genetics* 2019). In this context and in order to strengthen our initial observations, we have performed an entirely new experiment whose results are displayed in Supplementary Fig.19 and referred to in the revised manuscript lines 318-320. In this novel assay, we extended the HFSC feeding period (12 weeks instead of the 7.5 weeks used in our initial experiment) in order to promote greater collagen deposition and increased the number of animals involved (15-19 mice per group). The novel data again point to *Bnc2* heterozygosity leading to reduced liver collagen deposition and impaired HSC molecular activation (Supplementary Fig.19). Therefore, we feel our study provides strong evidences for a role of BNC2 in the control of HSC pro-fibrotic activation both *in vitro* and *in vivo*.

Reviewer #3 (Remarks to the Author):

The authors suggest a transcription factor, BNC2, as a novel transcriptional regulator in myofibroblastic activation in fibrosis. They combined computational analysis using public data as well as in-house genome-wide data to systematically assess and prioritize potential key players in fibrosis followed by *in vitro/in vivo* experimental validation. It was quite fun overall to read the manuscript and to see the incremental build-up of the story about *Bnc2*. The manuscript is well written, the evidence is convincing, and the conclusion has novelty and impact. A few methodological clarifications and additional mining of the genomewide data would make it publishable.

1. The authors used the notion of broad vs. sharp peaks of H3K4me3 to identify so-called “identity genes.” Although H3K4me3 is a representative promoter marker, H3K27ac is also enriched at promoter as well as enhancers. And superenhancers (SE) defined by H3K27ac are also known to represent the “key genes” of cell types. It is conceivable that SE analysis would capture *Bnc2* as well given the similarity of the H3K4me3 profiles and H3K27ac at the key gene locus, but FigS1B shows that H3K27ac displays both higher and broader enrichment. Can you perform SE analysis using H3K27ac and compare the associated genes with the currently identified genes by H3K4me3? If H3K27ac also captures a similar set of key genes, that would strengthen the gene prioritization; if not, that would also suggest distinct epigenetic roles in fibrosis, which would be also interesting on its own.

We concur with the reviewer that both broad H3K4me3 domains and super-enhancers can be used to predict identity genes. In fact, we and others have previously indicated that the two approaches point to partially common genes (Dubois et al. *Mol Syst Biol* 2020, Mikulasova et al. *Genome res* 2021). However, we have also recently described obvious limitations regarding the usage of super-enhancers to predict identity genes, which include shortcomings such as sensitivity to the enhancer stitching window used and issues related to target gene assignment (Dubois-Chevalier et al. *Epigenomics* 2020). Hence, we favored the approach based on broad H3K4me3 domains in this study. This rationale is stated in the revised manuscript lines 580-584. Nevertheless, we have defined super-enhancers using H3K27ac ChIP-seq as suggested by the reviewer. We used the seminal approach described in Whyte et al. *Cell* 2013 and Hnisz et al. *Cell* 2013, i.e. the enhancer stitching window was set at 12.5 kb and promoters were removed from the analysis to purely focus on enhancers. Due to aforementioned uncertainty about how to best assign target genes to super-enhancers, we did not link super-enhancers to specific genes but rather monitored the distance between the transcriptional start sites of genes associated with broad or sharp H3K4me3 domains to the nearest super-enhancer. This indicated that the genomic distribution of super-enhancers was significantly skewed towards genes labelled with broad H3K4me3 domains. These analyses also revealed a super-enhancer downstream

of the *BNC2* gene in MF-HSCs. Overall, these additional analyses confirm the epigenomic-based gene prioritization. These data are presented in novel panels in Supplementary Fig.1D and Supplementary Fig.3A and are referred to in the manuscript lines 111-112 and line 156. The methods and figure legends have also been modified accordingly.

2. In Fig S8B motif analysis:

a. Please include more detailed statistics including Hit%

Additional details about the de novo motif analysis are now provided in Supplementary Fig.9B as requested by the reviewer.

b. Any functional implications from the motif contents other than the similarity with previous publications?

The second motif identified in our *de novo* motif enrichment analysis presented in Supplementary Fig.9B is a GC-rich motif which could promote, among others, the binding of SMAD TFs (Martin-Malpartida et al. *Nat Commun* 2017). Additional motif enrichment analyses using TFmotifView (Leporcq et al. *Nucleic Acids Res* 2020), which searched for enrichment of motifs contained in the JASPAR database, identified the TEAD motif among the top hits. Altogether, these data are in line with our finding that the *BNC2* cistrome is largely shared with that of SMAD and TEAD/YAP1 (Fig.3B-C). Motif enrichment data are presented in Supplementary Fig.9B and C and mentioned in the manuscript lines 221-222. The Methods and figure legend of Supplementary Fig.9 have been amended to include a description of these analyses.

3. In Fig 3D, did the author test SMAD as well for co-IP?

In contrast to our data with YAP1, we could not detect an interaction between *BNC2* and SMAD in LX2 cells using Co-IP experiments. A representative result is shown here. Although we cannot rule out that a physical interaction between *BNC2* and SMAD might occur and be detectable in different experimental conditions, our result suggests that these transcription factors are connected through co-binding to shared cis-regulatory elements (Fig.3B-C and 3F-G) rather than through a direct interaction.

4. In Fig 4A&b:

a. What data are these? RNA-seq? or microarray? Method section contains very routine procedures only. Please clarify.

Transcriptomic data used in Fig.4A-B are from microarrays. This statement has been added to the figure legend. In order to limit the length of the Methods section, the main procedures and tools are described while more specific details are provided in the Supplementary Methods. Indications to refer to the Supplementary Methods are systematically provided when relevant throughout the Methods section of the manuscript.

b. What about down/up genes defined by FC or FDR in a canonical way? c. What are the overall characteristics of siBNC2-induced genes?

In order to answer those questions, genes differentially expressed between siBNC2 and siControl transfected LX2 cells were called and used for gene enrichment analysis using Metascape. Consistent with our GSEA analyses, down-regulated genes were specifically enriched for matrisome genes. Up-regulated genes were enriched for terms related to cellular stress response and RNA metabolism. These data are presented in a novel Supplementary Fig.13 and referred to in the manuscript lines 271-273.

5. In line 251: What does “deregulated” means? Does it imply both down and up-regulation?

Yes, both down- and up-regulated genes were considered in these analyses. This clarification has been added line 265.

6. In Fig 4A:

a. How exactly were these terms retrieved? Are these the top significant gene sets? Or fibrosis-related terms only?

First, the top enriched biological processes within genes bound by BNC2 in our LX2 ChIP-seq data were retrieved (Supplementary Fig.12A). Next, these terms were further grouped based on their similarity (Supplementary Fig.S12B and Supplementary Table S8) defining 6 main biological process classes, which are the ones displayed in Fig.4A. These main classes were used in Fig.4A to define whether associated genes were deregulated upon BNC2 silencing in LX2 cells. We have revised our description of these analyses by specifying that top hits were initially retained line 257 and by adding “Fig.4A” line 261 to clarify that terms in this panel are related to the ones identified in Supplementary Fig.S12B and Supplementary Table S8.

b. NES values are > 1. Does this mean that they are induced in siBNC2?

As indicated in response to point 5 (and now stated line 265), both up- and down-regulation was considered in these analyses. NES values > 1 indicate a biased distribution of genes of a given gene set towards those that are the most strongly deregulated (up or down) in siBNC2 transfected cells. This has been further clarified by showing results obtained with a random list of genes in Fig.4A, which serves as a reference for lack of specific enrichment in deregulated genes.

7. In Fig 4B:

a. How exactly is this different from Fig 4A except that it is “molecular function.” For any gene list, we can assess both the biological process and molecular function. Why only one in Fig 4A and the other in Fig 4B? b. Add statistics like Fig 4A if applicable.

In Fig.4B, we used a fraction of the genes from Fig.4A, i.e. only genes defined as both bound and regulated by BNC2. Indeed, while the GSEA data in Fig.4A indicate that genes contained in gene sets such as “Response to wounding – ECM” are enriched for deregulated genes in siBNC2 transfected LX2 cells, not all the genes from this gene set are actually modulated by BNC2 silencing. This is why Fig.4B only used the deregulated subset of genes from the gene sets analyzed in Fig.4A as stated lines 266-268. This rationale has been clarified by adding a schematic at the top of panels A and B and by providing greater details in the legend to Fig.4B. Fig.4B does not refer to GSEA data so no statistics similar to the ones shown in Fig.4A need to be presented in this panel.

8. In Fig 4I: One legend is missing for light green.

The legend of Fig.4I (now Fig.4K in the revised manuscript) has been checked to ensure no labels were missing.

REVIEWERS' COMMENTS

Reviewer #1 (Remarks to the Author):

The authors addressed all of my concerns.

Reviewer #3 (Remarks to the Author):

All comments have been addressed well. Recommended for publication.

Reviewer #4 (Remarks to the Author):

The authors describe the role of BNC2 in liver fibrosis and overall have a compelling story. As a new reviewer, i will focus primarily on the RIME experiments. The authors have adequately addressed the previous reviews 2's comments. The data is deposited, experimental details adequate and there are sufficient replicates. I have confidence in the BNC2-YAP1 interaction based on the number of RIME replicates, validation by CO-IP and co-binding on the genome at overlapping sites. If the authors have additional search results, such as peptide intensity scores etc, this could be added. It helps ranking proteins/peptides in the list. However, this is not essential. Further, like the authors have commented, linking direct peptide interactions from RIME is not an easy task.

Reviewer #1 (Remarks to the Author):
The authors addressed all of my concerns.

Reviewer #3 (Remarks to the Author):
All comments have been addressed well. Recommended for publication.

We would like to thank the reviewers for giving constructive criticism, which allowed to improve our manuscript.

Reviewer #4 (Remarks to the Author):

The authors describe the role of BNC2 in liver fibrosis and overall have a compelling story. As a new reviewer, I will focus primarily on the RIME experiments. The authors have adequately addressed the previous reviews' comments. The data is deposited, experimental details adequate and there are sufficient replicates. I have confidence in the BNC2-YAP1 interaction based on the number of RIME replicates, validation by CO-IP and co-binding on the genome at overlapping sites. If the authors have additional search results, such as peptide intensity scores etc, this could be added. It helps ranking proteins/peptides in the list. However, this is not essential. Further, like the authors have commented, linking direct peptide interactions from RIME is not an easy task.

We would like to thank the reviewer for appreciating our efforts to strengthen the RIME data. As suggested, we have added extra information to the supplementary data related to the RIME experiments (Supplementary data files 6 and 8) including identified peptide and protein confidence scores.